# Basal condensation of Numb and Pon complex via phase transition during *Drosophila* neuroblast asymmetric division

Zelin Shan[1], Yuting Tu[1], Ying Yang [2], Ziheng Liu[1], Menglong Zeng[3], Huisha Xu[4], Jiafu Long [4], Mingjie Zhang[3], Yu Cai [2] & Wenyu Wen [1]

Uneven distribution and local concentration of protein complexes on distinct membrane cortices is a fundamental property in numerous biological processes, including *Drosophila* neuroblast (NB) asymmetric cell divisions and cell polarity in general. In NBs, the cell fate determinant Numb forms a basal crescent together with Pon and is segregated into the basal daughter cell to initiate its differentiation. Here we discover that Numb PTB domain, using two distinct binding surfaces, recognizes repeating motifs within Pon in a previously unrecognized mode. The multivalent Numb-Pon interaction leads to high binding specificity and liquid-liquid phase separation of the complex. Perturbations of the Numb/Pon complex phase transition impair the basal localization of Numb and its subsequent suppression of Notch signaling during NB asymmetric divisions. Such phase-transition-mediated protein condensations on distinct membrane cortices may be a general mechanism for various cell polarity regulatory complexes.

[1] Department of Neurosurgery, Huashan Hospital, Institutes of Biomedical Sciences, State Key Laboratory of Medical Neurobiology, Department of Systems Biology for Medicine, School of Basic Medical Sciences, Shanghai Medical College of Fudan University, Shanghai 200032, China. [2] Temasek Life Sciences Laboratory, Department of Biological Sciences, National University of Singapore, Singapore 117604, Singapore. [3] Division of Life Science, State Key Laboratory of Molecular Neuroscience, Hong Kong University of Science and Technology, Clear Water Bay, Kowloon, Hong Kong, China. [4] State Key Laboratory of Medicinal Chemical Biology, College of Life Sciences, Nankai University, Tianjin 300071, China. Zelin Shan, Yuting Tu and Ying Yang contributed equally to this work. Correspondence and requests for materials should be addressed to Y.C. (email: caiyu@tll.org.sg) or to W.W. (email: wywen@fudan.edu.cn)

The ability to polarize is a fundamental property of most cells, touching on essentially all aspects of cellular processes[1,2]. Disruption in cell polarity can alter cell signaling and tissue structure to promote cancer formation and metastasis[3,4]. A hallmark of cell polarization is recruitment and prominent local condensation of certain protein complexes at specific membrane domains of a polarized cell[5–9]. Proteins within these complexes interact with each other forming highly concentrated assemblies (e.g., the conserved Par-3/Par-6/aPKC, Lgl/Dlg/Scribble, Prickle/Vangl, and Frizzled/Dishevelled/Diego complexes), which are peripherally associated with the inner surface of plasma membranes and are in open contacts with aqueous cytoplasm. Such asymmetric protein distribution is often highly dynamic and changes over time in response to cell intrinsic or extrinsic cues.

During the asymmetric cell division (ACD) of *Drosophila* neuroblasts (NBs), there are several unevenly distributed protein complexes: the Par-3/Par-6/aPKC complex and its related proteins Inscuteable, Partner of Inscuteable (Pins), and Gαi on the apical cortex; complexes of cell fate determinants and their adaptor proteins, including the Numb/Pon (Partner of Numb) complex; and the Prospero/Miranda complex on the basal cortex[7,9–11]. During the ACD of NBs, Pins orchestrates the formation of force generator to orient mitotic spindle along the apical-basal axis[12–15], leading to an unequal partitioning of fate determinants into the basal ganglion mother cell (GMC) daughter to initiate its differentiation. Intriguingly, both the apical and basal protein complexes each forms a crescent at the opposite pole of NB during ACD instead of uniformly distributed on either the apical or basal half of cortex[16–25]. Moreover, polarity proteins within the condensed crescents are in fast equilibrium with the cytoplasmic proteins[26–28]. It is not known how the highly concentrated protein crescents autonomously form in restricted membrane regions, and how the large concentration gradients between proteins within the crescent and in cytoplasm are maintained.

The basal localization of the Notch antagonist Numb[29–31] during ACD is regulated by the Par-3/Par-6/aPKC complex, its adaptor Pon, and cell cycle kinase Polo. aPKC-mediated phosphorylation precludes the apical cortex anchoring of Numb and Pon[32–34]. Meanwhile, Polo phosphorylates Pon, directing its basal recruitment together with Numb[18,35,36]. Both *polo* and *pon* mutant brains show over-proliferation of NBs in *Drosophila* larval central brain, and overexpression of Numb or introduction of *notch* RNAi could suppress the phenotype in the *polo* mutant, demonstrating that Polo functions upstream of the Numb-Notch pathway. Though the structural information illustrating Polo-Pon association and how Numb recognizes targets through its PTB domain is available[35,37,38], the molecular basis of Pon-mediated Numb localization is not clear yet. Moreover, evidence for a direct role of Pon in inhibiting NB self-renewal via the Numb-Notch signaling pathway is lacking, and no detailed study demonstrating its association with Numb in the regulation of Polo-Pon-Numb-Notch axis exists.

In this work, we discover that Numb PTB specifically recognizes repeating motifs located at the N terminus of Pon in a non-canonical fashion. The crystal structures of Numb PTB domain in complex with its binding regions of Pon reveal that the Numb/Pon interaction is govern by highly specific, multivalent interaction between the two proteins, which further leads to liquid-liquid phase separation (LLPS) of the complex both in vitro and in living cells. The Numb/Pon complex in the condensed liquid phase is highly concentrated and rapidly exchanges with the two proteins in the aqueous phase of the cytoplasm. Moreover, we provide evidence that Pon functions upstream of the Numb-Notch axis to promote larval NB differentiation, and the multivalent Numb/Pon interaction is crucial for Pon function during ACD of neural stem cells. Pon mutants that are not capable of forming phase transition with Numb lose its ability to direct Numb basal localization, have impaired Notch inhibition by Numb, and result in tumor-like over-proliferation of *Drosophila* NBs. Our study suggests that multivalent protein-protein interaction-induced phase transitions may be a general mechanism for condensing various polarity regulatory complexes in distinct membrane cortices in polarized cells.

## Results

**Numb PTB recognizes repeating motifs in N terminus of Pon.** We first confirmed the specific interaction between Numb PTB and an N-terminal fragment (amino acids (aa) 1–228) of Pon (Supplementary Fig. 1a, b and Supplementary Table 1). Interestingly, the relative amount of Numb PTB pulled down by GST-Pon1-228 was much greater than that of Pon1-228 pulled down by GST-Numb PTB (Supplementary Fig. 1c, d), suggesting the existence of multiple PTB-recognition motifs in Pon1-228.

Several repeating motifs have been found in Pon1-228: type A "FxNxx[F/L]" motif and type B "NP[F/Y]E[V/I]xR" motif (Fig. 1a). Proper combination of both motifs dramatically increased the interaction with Numb PTB, whereas the isolated one barely bound to it (Fig. 1b–d and Supplementary Fig. 2a). For example, Pon fragments containing either B1A2, A2B2, or A3B3 showed significantly enhanced binding avidity to PTB compared with the isolated A or B motifs. In contrast, A1B1 weakly interacted with Numb PTB, and B2A3 did not bind to PTB at all. Moreover, combination of more coupled AB motifs, such as A2B2A3B3 (referred to as A2B3 hereafter), or even A1B1A2B2A3B3 (Pon1-228, referred to as A1B3 hereafter) significantly strengthened the Numb-Pon interaction (Fig. 1d), suggesting that multiple copies of AB repeats could act cooperatively for high avidity PTB binding.

**Structure of the Numb PTB and Pon complexes.** To elucidate how combination of A and B motifs can strengthen Pon's binding to Numb PTB, we solved the crystal structures of Numb PTB in complex with Pon A2B2 or Pon B1A2 peptides (Table 1). The complex structures revealed that either Pon A2B2 or Pon B1A2 binding led PTB to form a symmetric dimer-of-dimers in either a "head-to-head" or "tail-to-tail" organization (Fig. 2a, b and Supplementary Fig. 3a–c). Pon binding could also induce Numb dimerization in solution (Supplementary Fig. 3d, e). Despite of the distinct domain arrangement in the two complexes, the A or B motifs from B1A2 and A2B2 occupied the identical sites on PTB (Fig. 2a, b and Supplementary Fig. 3f, g). The A motif adopts a similar fold as the canonical PTB-binding "NPxY"/"NxxF" motif, with its key Asn139$^{A2}$ forming hydrogen bonds with PTB[39]; the B motif binds to PTB through extensive hydrophobic and polar interactions (Fig. 2c and Supplementary Fig. 3h). For example, the bulky aromatic ring of Tyr154$^{B2}$ (same as Phe120$^{B1}$) inserts into a hydrophobic pocket formed by the side chains of Tyr87, Val117, and Ile139 from PTB. The side chain of Arg158$^{B2}$ (same as Arg124$^{B1}$) forms hydrogen bonds with the backbone carbonyl oxygen of Glu92$^{PTB}$. Importantly, these key residues involved in PTB binding are highly conserved in the A and B motifs (Fig. 1a).

In line with the structural data, mutations in either the A site (G192L$^{PTB}$, G192D$^{PTB}$, N139A$^{A2}$, and N169A$^{A3}$) or B site (C90W$^{PTB}$, R171A$^{PTB}$, E121A$^{B1}$, N152,P153A$^{B2}$, E155A$^{B2}$, Y154E$^{B2}$, R158A$^{B2}$, F186E$^{B3}$, and E187A$^{B3}$) completely destroyed or significantly impaired the interaction between Numb PTB and Pon fragments containing only one copy of AB combination

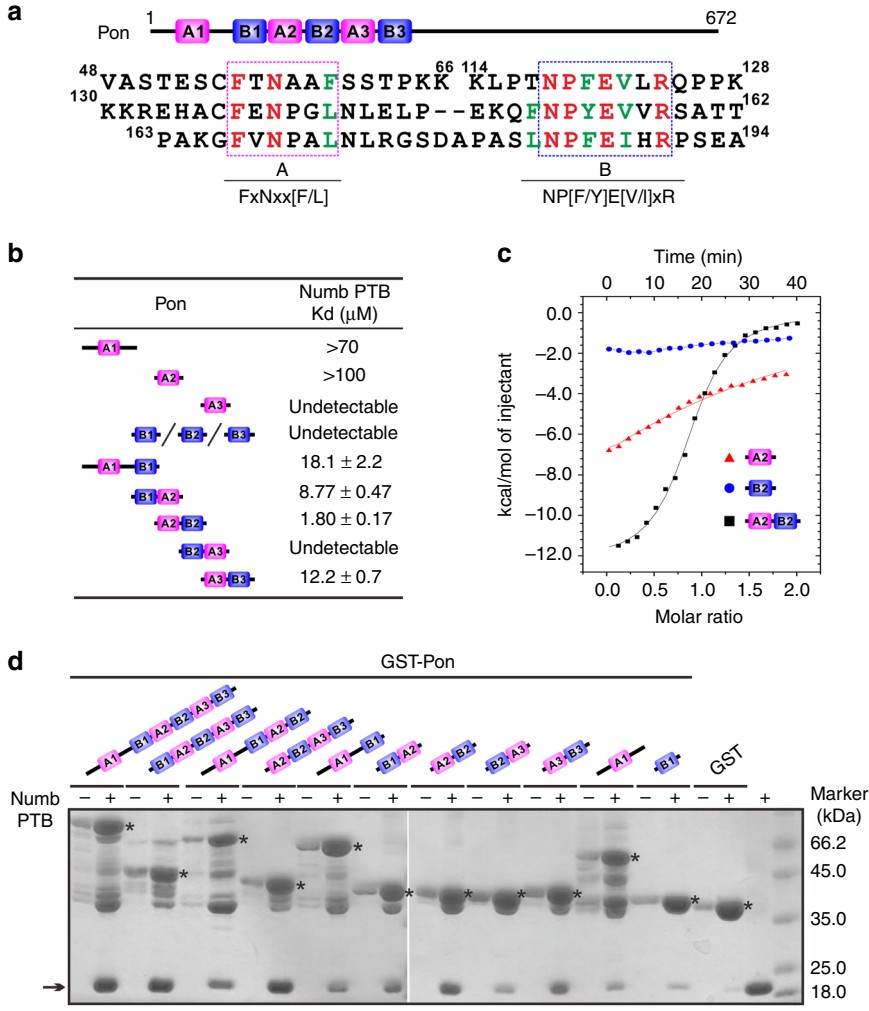

**Fig. 1** Pon binds to Numb PTB with combined AB motif repeats. **a** Existence of repeated A and B motifs in Pon. The A and B motifs are indicated with dashed magenta and blue boxes, respectively. The completely conserved residues were highlighted in red, and the highly conserved residues are in green. **b** Summary of the quantitative binding constants between various Numb PTB and Pon fragments derived from the ITC-based titration assays shown in Supplementary Fig. 2a. **c** ITC-based measurements of the binding between Numb PTB and Pon A2B2, Pon A2, or Pon B2 fragments. **d** GST pull-down assay showing that repeating AB motifs enhanced the binding avidity of Pon to Numb PTB. Numb PTB is indicated with an arrow, and GST-Pon fragments are indicated with asterisks. Uncropped gels are shown in Supplementary Fig. 10

(Fig. 2c, d and Supplementary Figs. 2b-d, 3h and 4a, b). Whereas for Pon full-length protein or fragments containing multiple copies of AB repeats, mutations of single copy of AB repeat only weakened the Numb-Pon interaction to certain extent, and combined mutations of all the AB repeats, e.g., the tetra- (N139A, Y154E and N169A,F186E, referred to as 4M hereafter) and hexa- (N57A,F120E, N139A,Y154E, and N169A,F186E, referred to as 6M hereafter) mutations, completely destroyed the interaction (Fig. 2e and Supplementary Fig. 4c–e). As expected, either C90W[Numb] or G192D[Numb] mutation severely impaired the Numb-Pon interaction (Fig. 2f).

**A potential common mode for target recognition of PTBs.** The bidentate-binding mode was also observed when Numb interacted with Numb-associated kinase (Nak). Amino-acid sequence analysis of Nak revealed the presence of a type B-like motif after the type A "NxxF" motif (Fig. 3a). Inclusion of the B motif significantly enhanced the binding of Nak to Numb PTB (Fig. 3b). In contrast to Pon A2B2, the Nak AB motif binding led to a 1:1 PTB/Nak heterodimer formation in solution, implying that the Nak AB motifs bind to the two sites within the single Numb PTB

molecule (Fig. 3c). Note that the linker region connecting the A and B motifs in Nak is long enough for the Nak A and B motifs to occupy the two binding sites on the same PTB.

By analyzing all the PTB or PTB/ligand structures deposited in the Protein Data Bank, we further found that the newly identified B motif-binding hydrophobic pocket on the *Drosophila* Numb PTB appears to be present in most of other PTB domains (Supplementary Fig. 5). We then evaluated the binding specificity enhancement provided by the B site using the interaction between the Pon A2B2 motif and two highly homologous PTB domains: mouse Numb PTB; and rat MINT2 PTB. Both of these PTB domains have been shown to interact, albeit weakly, with proteins containing the type A "NPxY"/"NxxF" fragments[39]. However, we could not detect binding between either of these two PTB domains and Pon A2B2, possibly due to the variation of B sites on these PTBs (residue Val108 in fly Numb PTB is replaced by Arg in both mouse Numb PTB and rat MINT2 PTB, Fig. 3d). Presumably, the long side chain of Arg would have severe steric hindrance with the main chain of B motif if it resides on the B motif-binding pocket, suggesting that proper utilization of B motif would significantly increase the target-binding selectivity of PTB domains (Fig. 3e).

**Table 1 Data collection and refinement statistics**

|  | Numb PTB/Pon A2B2 complex | Numb PTB/Pon B1A2 complex |
|---|---|---|
| *Data collection* |  |  |
| Space group | P3221 | P3112 |
| Cell dimensions |  |  |
| *a, b, c* (Å) | 43.392, 43.392, 181.304 | 65.593, 65.593, 143.879 |
| α, β, γ (°) | 90, 90, 120 | 90, 90,120 |
| Resolution (Å) | 50.00–2.00 (2.03–2.00)ᵃ | 50.00–1.70 (1.73–1.70)ᵃ |
| $R_{merge}$ (%) | 5.2 (76.2) | 5.6 (79.5) |
| $I/\sigma I$ | 24.6 (2.0) | 52.1 (3.1) |
| Completeness (%) | 98.6 (96.7) | 99.3 (97.7) |
| Redundancy | 4.8 (4.3) | 9.8 (9.4) |
| *Refinement* |  |  |
| Resolution (Å) | 36.80–2.00 | 30.39–1.70 |
| No. of reflections | 13,901 | 39,079 |
| $R_{work}/R_{free}$ | 19.1/22.7 | 19.4/21.8 |
| No. of atoms |  |  |
| Protein | 1222 | 2164 |
| FMT | 3 |  |
| Glycerol |  | 12 |
| Water | 53 | 155 |
| *B*-factors |  |  |
| Protein | 30.84 | 29.00 |
| FMT | 41.56 |  |
| Glycerol |  | 38.18 |
| Water | 31.58 | 32.91 |
| R.m.s. deviations |  |  |
| Bond lengths (Å) | 0.010 | 0.011 |
| Bond angles (°) | 0.951 | 1.138 |

ᵃ Values in parentheses are for highest-resolution shell

**Phase transition of the Numb and Pon complex in vitro.** Interestingly, by mixing Numb PTB and Pon A1B3 under light microscope, we observed numerous small spherical droplets with various diameters, a phenomenon characteristic of LLPS (Fig. 4a). These small droplets gradually fused into larger ones within minutes (Fig. 4b and Supplementary Movie 1). In sharp contrast, isolated solutions of Numb PTB and Pon A1B3 remained clear at the same concentration (Fig. 4a).

We separated the condensed liquid phase from the bulk aqueous solutions by centrifugation[40], and found that LLPS of the Numb PTB/Pon A1B3 complex is concentration-dependent (Fig. 4c, d and Supplementary Fig. 6a). For the 1:1 molar ratio mixture of Numb PTB and Pon A1B3 at 25 μM, approximately one-third of Pon A1B3 and two-thirds of Numb PTB were recovered from the condensed liquid phase (Fig. 4c, d). More Numb PTB and Pon A1B3 (~80%) could be recovered from the condensed liquid phase when the sample concentrations increased to 100 μM (Supplementary Fig. 6a). Neither Numb PTB nor Pon A1B3 alone could form condensed liquid phase at up to 100 μM concentrations. Considering the large volume differences between the condensed liquid droplet pellet and the supernatant aqueous solution, protein concentrations for both Numb PTB and Pon A1B3 in the condensed liquid phase are much higher (estimated to be at least 100-fold higher) than those in the aqueous solution.

Next, we found that the multivalency of Numb-Pon interaction is required for phase separation. Single copy of AB motifs (e.g., A1B1, B1A2, A2B2, and A3B3) or A2B3 cannot cause LLPS by binding to Numb PTB; condensed lipid phase only formed between Pon A1B3 or A1B2 and Numb PTB, though the latter is less efficient (Fig. 4a–c and Supplementary Fig. 6b, c and

Supplementary Movie 2). We further noted that Numb/Pon LLPS is molar ratio-dependent, and almost all the proteins (>90%) shifted to the condensed liquid phase in a 1:3 molar ratio mixture of Pon A1B3 and Numb PTB (Supplementary Fig. 6d), a phenomenon well correlated with the existence of three copies of AB motifs within Pon (Fig. 1a). Consistent with these data, mutations on Numb PTB or Pon A1B3 that disrupts the Numb-Pon binding blocked Numb/Pon LLPS in a 1:1 Pon/Numb mixture (Supplementary Fig. 6e), whereas Pon N169A,F186E, Pon N57A,F120E, Pon N139A,Y154E, Pon 4M, and Pon 6M are progressively less effective than the Pon wild type (WT) in driving LLPS in a 1:2 Pon/Numb mixture (Fig. 4e). Moreover, the Numb/Pon complex LLPS can be reversed by a monovalent competing Numb PTB ligand. Addition of a pPon peptide[35], which binds to the A site of Numb PTB (Kd ~11 μM, Supplementary Fig. 2f), led to immediate dispersion of the pre-formed Numb/Pon liquid phase into homogenous aqueous solution (Fig. 5a and Supplementary Fig. 6f and Supplementary Movie 3) in a peptide concentration-dependent manner (Fig. 5b). Furthermore, addition of 1,6-hexanediol, an aliphatic molecule that is reported to disturb hydrophobic interaction-induced phase separation assemblies both in vitro[41] and in vivo[42,43], also led to dispersion of the pre-formed Numb/Pon liquid droplets (Fig. 5c) in a concentration-dependent manner (Fig. 5d), further demonstrating that the condensed Numb/Pon liquid droplets are LLPS assemblies.

**Phase transition of the Numb and Pon complex in living cells.** We have also observed LLPS and time-dependent fusion into larger liquid droplets of the Numb/Pon complex using iFluor^TM 488-labeled Pon A1B3 and Cy3-labeled Numb PTB by fluorescence microscopy (Fig. 6a and Supplementary Fig. 7a and Supplementary Movie 4). Fluorescence recovery after photobleaching (FRAP) analysis of Cy3-Numb PTB droplets demonstrated that Numb PTB molecules constantly exchange between droplets and the surrounding aqueous solution (Fig. 6b and Supplementary Movie 5).

Moreover, the Numb/Pon complex can autonomously assemble into highly concentrated, membrane-lacking nuclear compartments/puncta in living cells. When GFP-Pon A1B3 and Cherry-Numb PTB were co-expressed in HeLa cells, we observed many bright puncta in the nucleus containing both green fluorescent protein (GFP) and Cherry signals (Fig. 6c). No puncta were observed in cells when only Pon A1B3 or Numb PTB was expressed (Supplementary Fig. 7b). Nicely correlated with the in vitro phase transition experiment shown in Fig. 4e, no or very little puncta could be detected when Numb PTB was co-expressed individually with Pon A1B3 mutants N139A,Y154E, 4M, and 6M (Fig. 6d). Similarly, co-expression of a Numb PTB mutant (C90W,G192D) with WT Pon A1B3 also eliminated the puncta formation. Whereas when Numb PTB and Pon A1B3 N169A, F186E mutant were co-expressed, a decreased but significant number of puncta could be detected (Fig. 6d). These data indicate that the Numb PTB- and Pon A1B3-enriched puncta formation also depends on the specific and multivalent interaction between the two proteins.

We noted that GFP-Pon in the puncta is also in a rapid dynamic equilibrium with the surrounding protein (~80% FRAP recovery with a half-time ~5 s; Fig. 6e, f and Supplementary Movie 6), a phenomenon reminiscent of the dynamic association of Numb/Pon with the crescent cortex during ACD of NBs and sensory organ precursor (SOP) cells[27,28], implying that the polarized distribution of Numb might be induced by Numb/Pon complex-mediated LLPS in asymmetrically dividing *Drosophila* NBs and SOP cells.

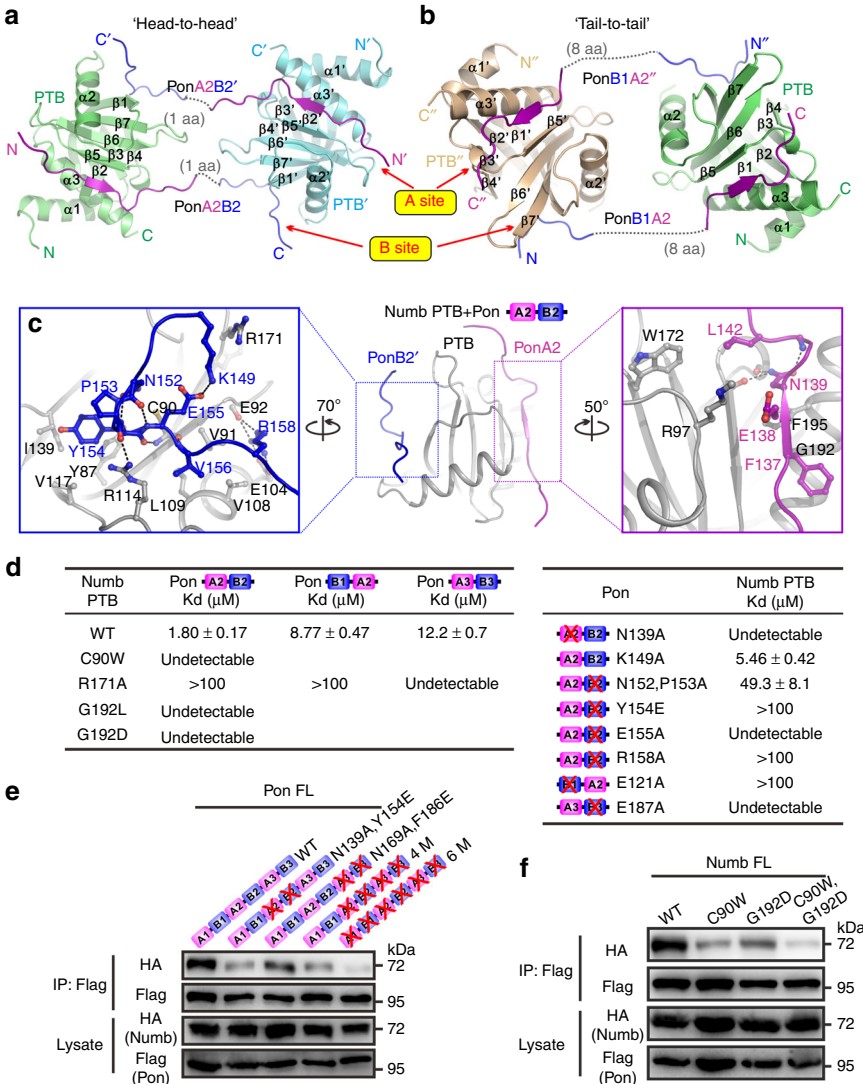

**Fig. 2** The crystal structures of Numb PTB in complex with Pon A2B2 or Pon B1A2. **a, b** Ribbon representations of Numb PTB/Pon A2B2 (**a**) and Numb PTB/Pon B1A2 (**b**) complexes as viewed from the top. The A and B sites are indicated. **c** The interaction details between the representative Numb PTB (gray) and Pon A2B2 (purple and light yellow) complex. Charge-charge and hydrogen-bonding interactions are highlighted by dashed lines in black. **d** Summary of the quantitative binding constants between various Numb PTB and Pon fragments derived from the ITC-based titration assays shown in Supplementary Fig. 2b-d. **e** In HEK293T cells, N139A,Y154E$^{Pon}$, N169A,F186E$^{Pon}$, 4M$^{Pon}$, or 6M$^{Pon}$ mutation abolished or significantly impaired the interaction between Numb and Pon. **f** Numb$^{C90W}$, Numb$^{G192D}$, and Numb$^{C90W,G192D}$ could not coimmunoprecipitate with Pon$^{WT}$. Uncropped blots are shown in Supplementary Fig. 10

**Binding to Pon is required for Numb localization in vivo**. To investigate the functional significance of the Numb-Pon interaction, we employed type I NBs in the central brain region of *Drosophila* larvae (Fig. 7a) as an in vivo model to examine the behavior of these molecules. To this end, we generated transgenic flies expressing full-length Flag-tagged WT or mutant forms of Numb and Pon, respectively, to address their subcellular localization.

During NB division, Pon together with Numb is basally localized and is segregated into the basal daughter cell[18,27,29,30,36]. We first examined the localizations of these transgenes in a WT background and focused on dividing NBs. Like Flag-Pon WT, Flag-Pon N139A,Y154E, Flag-Pon N169A,F186E, and Flag-Pon 4M also localized on the basal cortex in metaphase NBs ($n = 20$, 100%; Supplementary Fig. 8a, c), implying the N-terminal fragment of Pon does not play a key role for Pon basal targeting in a WT background and is consistent with previous finding showing that the C-terminal localization domain (aa 493–672)

directs its basal localization[27]. In line with this, the localization of endogenous apical complex component aPKC and basal cell fate determinant Numb and adaptor protein Miranda (Mira) in these backgrounds was normal (Supplementary Fig. 8a). Predictably, Flag-Numb WT ($n = 20$, 100%) showed the expected basal enrichment in metaphase NBs (Supplementary Fig. 8b, d). In contrast, the Pon-binding deficient Flag-Numb C90W ($n = 20$, 100%) was largely cytoplasmic with weak basal localization although endogenous Pon (as well as other asymmetric proteins, including Mira and aPKC) was localized on basal (or apical for aPKC) cortex (Supplementary Fig. 8b), supporting our structural analysis and in vitro results and, demonstrating that the direct interaction between Pon and Numb PTB is responsible for the correct localization of Numb during ACD.

**Numb localization requires its multivalent binding to Pon**. It is worth noting that endogenous Numb in NBs ectopically expressing Pon variants that disrupt Pon-Numb binding, remained its

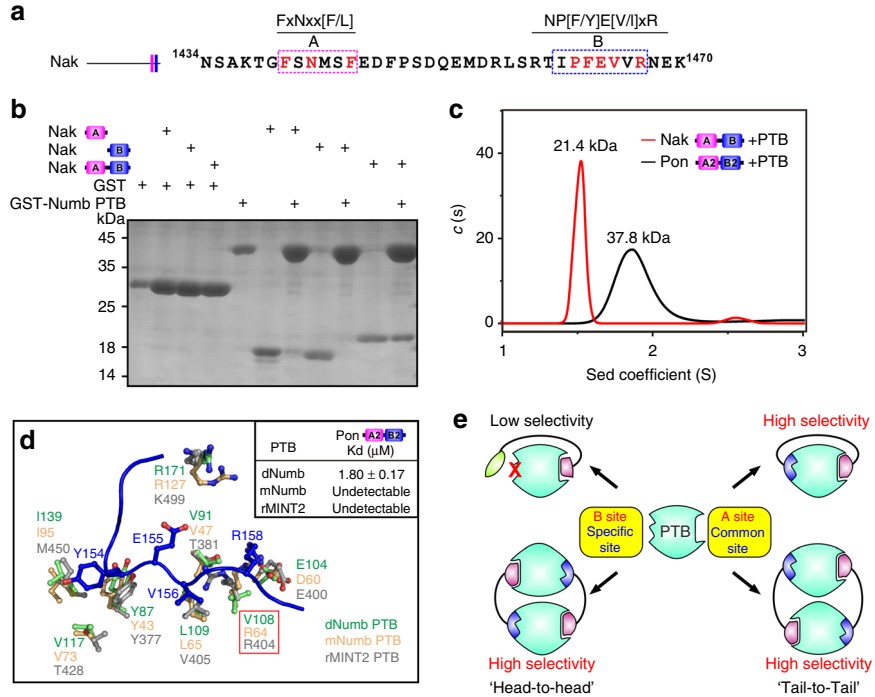

**Fig. 3** Efficient combination of A and B motifs enhances the target-binding affinity and selectivity of PTB domain. **a** The C terminus of Nak contains both A and B motifs. **b** GST pull-down assay showing that combination of both A and B motifs strengthened the interaction between Nak and Numb PTB. The uncropped gel is shown in Supplementary Fig. 10. **c** Sedimentation velocity (SV) experiments of Numb PTB in the presence of three molar ratios of Nak AB. The SV profile of PTB/Nak AB displays as a major peak corresponding to a 1:1 heterodimer (21.4 kDa, red line). The theoretical molecular weights of PTB and Nak AB are 15.7 and 4.0 kDa, respectively. The SV profile of Numb PTB/Pon A2B2 is also shown for the reference (black line). **d** Superimposition of *Drosophila* Numb PTB/Pon A2B2 complex (green) with mouse Numb PTB (orange, PDB ID: 1WJ1) and rat MINT2 PTB (gray, PDB ID: 3SV1). Pon A2B2 selectively binds to *Drosophila* Numb PTB, but not mouse Numb PTB or rat MINT2 PTB. The key residues involved in Pon B2 binding are shown in the ball-and-stick model. **e** A model depicting potent and specific PTB-target recognitions

basal localization (Supplementary Fig. 8a). We reasoned that the correct basal localization of endogenous Numb is mediated by endogenously expressed Pon. To investigate this, we next addressed endogenous Numb localization in *pon* mutant NBs rescued with these Pon variants. As reported previously[27], Numb was basally co-localized with Pon in WT NBs (Fig. 7b, $n = 20$, 100%) but uniformly localized on the whole cortex (as well as some cytoplasmic distribution) in *pon* mutant NBs (Fig. 7c, $n = 17$, 100%). Expression of Flag-Pon WT in *pon* mutant NBs restored the basal localization of Numb ($n = 11$, 82%) in the majority of dividing NBs (Fig. 7d, e). Similar to Flag-Pon WT ($n = 11$, 100%), Flag-Pon N139A,Y154E ($n = 12$, 100%), Flag-Pon N169A,F186E ($n = 12$, 100%), and Flag-Pon 4M ($n = 13$, 100%) also formed basal crescent in *pon* mutant NBs (Fig. 7d), confirming that the AB repeats at the N-terminal region of Pon is not required for its basal targeting. However, the basal localization the endogenous Numb was only partially restored ($n = 12$, 50% for Flag-Pon N139A,Y154E; $n = 12$, 75% for Flag-Pon N169A,F186E; and $n = 13$, 46.1% for Flag-Pon 4M) in these backgrounds compared with that of the Flag-Pon WT (Fig. 7d, e), pointing to the importance of these identified residues in directing the correct basal localization of Numb. As an internal control, Mira and aPKC were normally localized in *pon* mutant NBs with or without expressing of different Flag-Pon variants (Fig. 7c, d; $n = 10$, 100%).

To further understand the function of these Flag-Pon variants, we investigated whether they could rescue *pon* mutant clone over-proliferating phenotype. In the WT central brain, each type I NB lineage contained one cell expressing NB marker Dpn ($n = 31$, 100%), however majority of *pon* mutant NB clones contained multiple Dpn-positive cells (12.1, $n = 26$, Fig. 8a–c, g). Similarly, type I *pon* mutant NB in ventral nerve cord (VNC) region also

contained more Dpn-positive cells (1.87, $n = 23$ in VNC, Supplementary Fig. 9a, b), which also expressed Asense (Ase), another NB marker. While restoring Flag-Pon WT expression in *pon* mutant NBs largely reverted the formation of ectopic Dpn-positive cells, ectopic expression of Numb-binding deficient Flag-Pon N139A,Y154E and Flag-Pon 4M did not rescue NB over-proliferative phenotype in *pon* mutant, and Flag-Pon N169A, F186E only partially suppressed this phenotype (Fig. 8c, g), showing that proper Numb-targeting is important for Pon function. The rescuing efficiency of endogenous Numb basal localization and NB over-proliferative phenotype in *pon* mutant NBs by various Flag-Pon mutants was nicely correlated with their abilities to induce LLPS through Numb binding in vitro and in heterologous cells (Figs. 4e, 6d, 7e, and 8g and Supplementary Table 1), implying that the multivalent Numb-Pon interaction-induced LLPS is critical for the efficient Numb enrichment in the basal crescent and Pon function. Further supporting the above conclusion, larval brain treated with 1,6-hexanediol, which can disturb Numb/Pon LLPS assemblies in vitro (Fig. 5c, d), exhibited defective localizations of both endogenous Numb and Pon in a concentration-dependent manner (Fig. 9a, c). In these NBs, while Pon localized in cytoplasm, Numb was cortically localized with weak cytoplasm, reminiscent of its localization in *pon* mutant NB. Importantly, removal of 1,6-hexanediol restored the formation of Numb and Pon crescents in dividing NBs (Fig. 9b, c), demonstrating that the dynamic basal condensation of Numb/Pon is mediated by LLPS.

**Pon functions upstream of the Numb and Notch pathway.** We also investigated whether the Pon-binding defective Numb variant could rescue *numb* mutant clone over-proliferating

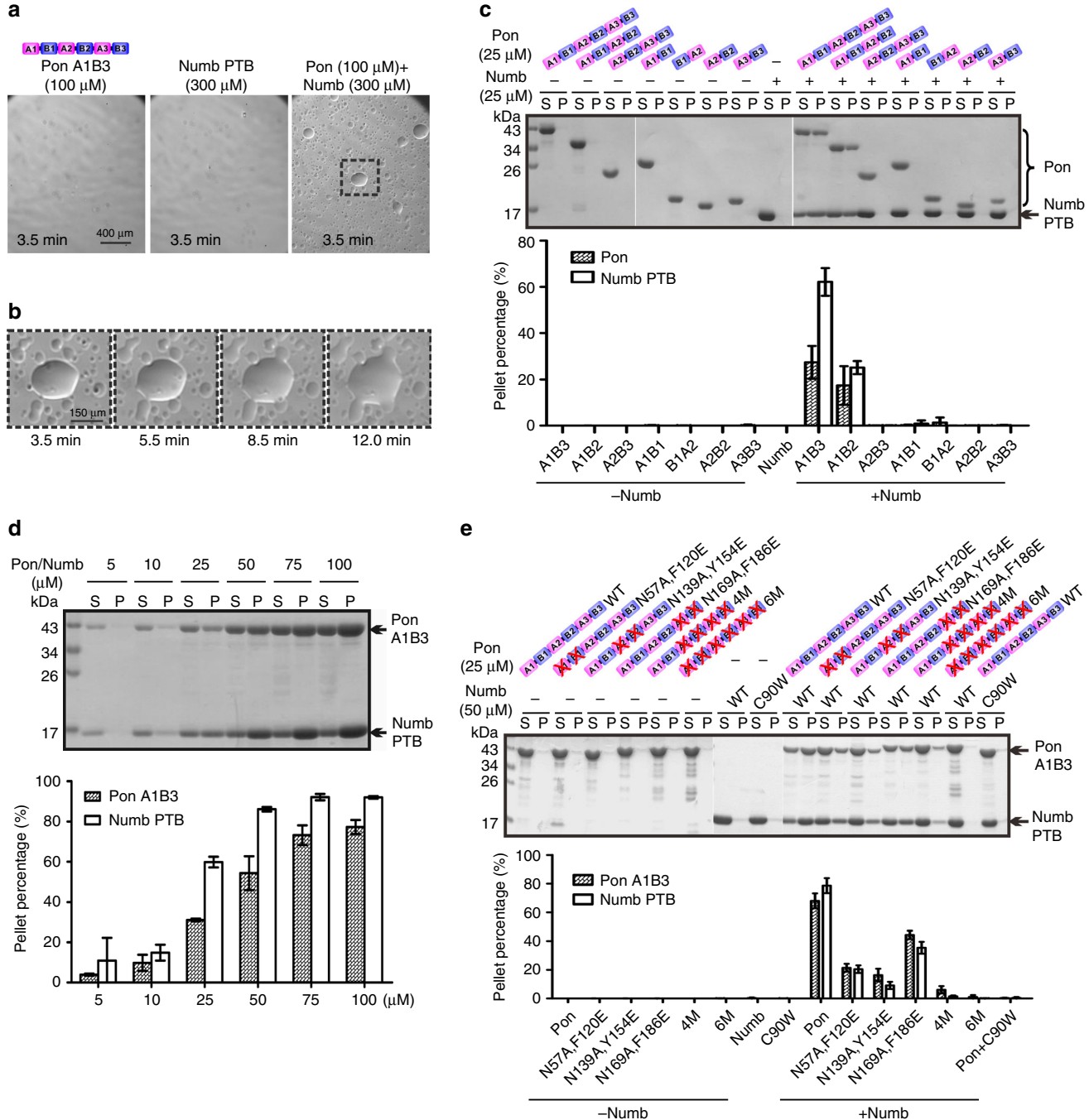

**Fig. 4** Phase transition of the Numb PTB/Pon A1B3 complex. **a** Isolated Numb PTB and Pon A1B3 solutions are clear under light microscope at room temperature (RT). Mixing the two proteins led to formation of numerous droplets (see also Supplementary Movie 1). The images were acquired 3.5 min and onward after mixing. The dashed box is the region of zoomed-in analysis in **b**. **b** The small droplets underwent time-dependent fusion into larger ones. **c** Representative SDS-PAGE analysis and quantification data showing the distribution of proteins between aqueous solution/supernatant (S) and condensed liquid droplets/pellet (P) fractions for various Numb PTB/Pon fragment mixtures. Compared with Pon A1B3, Pon A2B3 is less efficient to form liquid droplets by binding to Numb PTB, whereas other Pon fragments cannot form liquid droplets by binding to Numb. **d** Sedimentation assay showing that the phase transition of the Numb PTB/Pon A1B3 complex is concentration-dependent. Numb PTB and Pon A1B3 were mixed at a 1:1 molar ratio with final concentration indicated. **e** Numb-Pon-binding deficient Pon A1B3 mutants (N57A,F120E, N139A,Y154E, N169A,F186E, 4M, and 6M) or Numb PTB mutant (C90W) showing significantly impaired phase transition when compared to the WT proteins. All statistic data in this figure represent the results from three independent batches of experiments and are expressed as mean ± SD. Uncropped gels are shown in Supplementary Fig. 10

phenotype. Consistent with previous report[36], we found more than half of *numb* mutant NB clones (56%, $n = 50$) overproliferated and contained multiple Dpn- and Ase-positive cells in the central brain, which is in sharp contrast to WT control clones (Fig. 8d, e). While introducing Flag-Numb WT in *numb*

mutant NBs largely reverted the formation of ectopic Dpn- and Ase-positive cells, ectopic expression of Flag-Numb C90W could not suppress this phenotype, reinforcing that proper targeting by Pon via its AB motif-mediated binding to Numb PTB is important for Numb function.

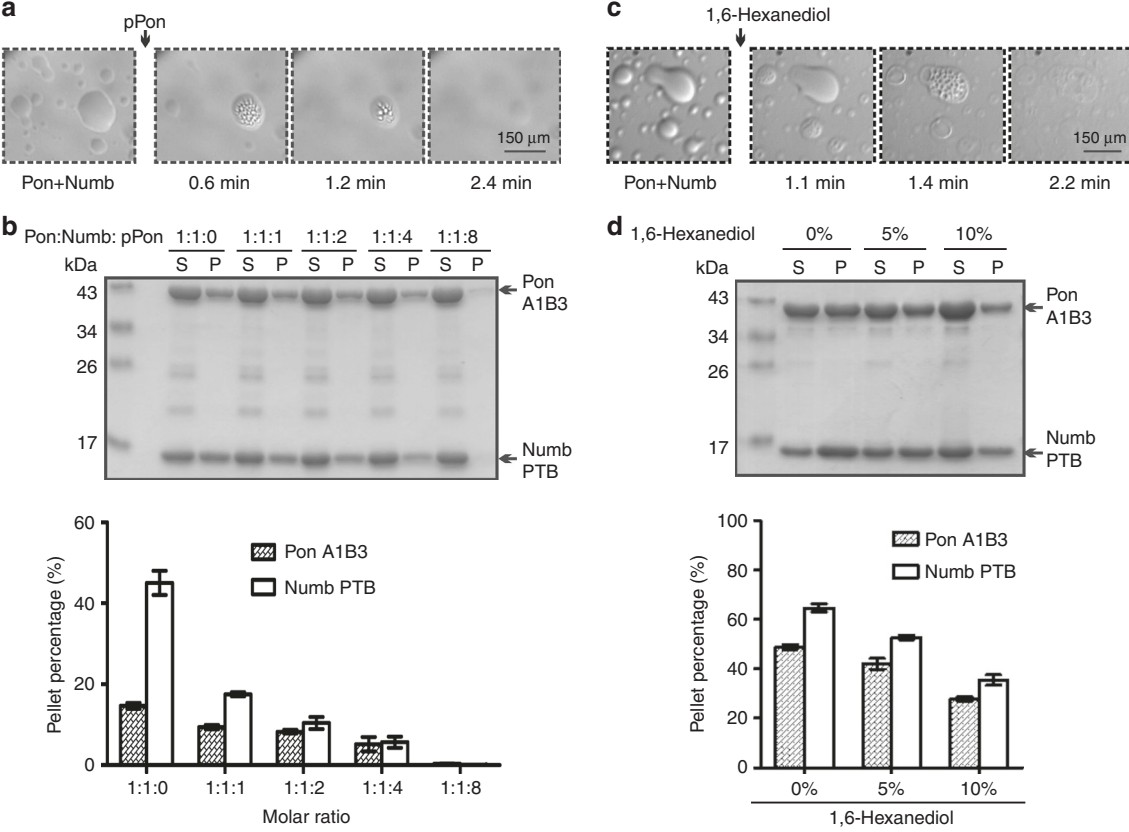

**Fig. 5** The Numb/Pon phase separation assemblies can be disturbed. **a** Pre-formed Numb PTB (150 μM)/Pon A1B3 (50 μM) droplets are rapidly dispersed after adding the pPon peptide (see also Supplementary Movie 3). The arrow refers to the time point of adding the pPon peptide to the mixture. **b** Pre-formed Numb PTB (20 μM)/Pon A1B3 (20 μM) droplets can be reversed to aqueous phase by the competing pPon peptide in a dose-dependent manner. The molar ratios of Pon A1B3, Numb PTB, and pPon are indicated. **c** Pre-formed Numb PTB/Pon A1B3 (60 μM) droplets are rapidly dispersed after adding 10% 1,6-hexanediol. The arrow refers to the time point of adding 1,6-hexanediol to the mixture. **d** Pre-formed Numb PTB/Pon A1B3 (60 μM) droplets can be reversed to aqueous phase by 1,6-hexanediol in a dose-dependent manner. The final concentration of 1,6-hexanediol is indicated. All statistic data in this figure represent the results from three independent batches of experiments and are expressed as mean ± SD. Uncropped gels are shown in Supplementary Fig. 10

Pon is required for the asymmetric segregation of Numb into the GMC daughter, where Numb inhibits Notch activity by regulating its degradation to drive GMC daughter toward differentiation process[18,29,30,36,44]. Consequently, *numb* mutant NBs exhibit elevated Notch expression and it was proposed that *numb* mutant clone over-proliferates in a Notch-dependent manner[44]. To address this, we conducted genetic interaction by knocking down Notch in *numb* mutant clone and found that introduction either one of two independent *notch* RNAi constructs efficiently reverted *numb* mutant clone over-proliferative phenotype (Fig. 8d, e and Supplementary Fig. 9c, d). These results support the notion that Numb promotes GMC daughter differentiation by inhibiting the Notch function. It is worth noting that NBs expressing these *notch* RNAi constructs exhibited normal localization of both apical and basal proteins (Supplementary Fig. 9e).

Next, we tested whether the observed more NB-like cells in *pon* mutant clone is also a result of elevated Notch signaling activity. In line with the notion that Pon-mediated basal localization of Numb is important for its activity, ectopic expression of either Flag-Numb WT or Flag-Numb C90W in *pon* mutant NBs could not revert the formation of ectopic Dpn- and Ase-positive cells. Interestingly, reducing Notch in *pon* mutant NBs strongly suppressed the over-proliferative phenotype (Fig. 8f, g and Supplementary Fig. 9f). Together, these results indicate that Pon may fill in the gap between cell cycle kinases Polo and

Numb-Notch signaling pathway in regulating ACD, and the proper targeting and local concentration of Numb by Pon on the basal cortex is essential for its subsequent inhibition of Notch signaling.

## Discussion
During development, a limit number of neural stem cells give rise to many different types of neurons and glia via ACDs. As the first identified cell fate determinant, Numb transiently forms a basal crescent during mitosis, preferentially segregates to the basal GMC daughter, and then promotes its differentiation to neurons/glia by antagonizing Notch signaling. We found that Numb PTB specifically recognizes the AB motif repeats of its adaptor Pon in a previously unrecognized manner. The multivalent interaction between Numb and Pon can lead to LLPS of the complex, forming condensed, autonomously assembled membrane-lacking compartments both in vitro and in living cells. Both Numb and Pon are highly concentrated in the condensed phase of the mixture. The pre-formed condensed phase droplets can be reversed by a monovalent competing Numb PTB ligand. As the Numb/Pon assemblies are attached to the basal cortex in dividing NBs, possibly through their membrane-binding domains or the third basal anchoring protein[27,45], it is supposable that the round Numb/Pon phase droplets seen in vitro and in Hela cells could be mechanically pulled into the cap shape (crescent from the side

view) in dividing NBs. Importantly, mutations that disrupt efficient LLPS of the Numb/Pon complex led to diffusion of Numb on the cortex during *Drosophila* NB division, and consequently resulted in ACD defects and tumor-like over-proliferation of NBs, presumably due to impaired Notch inhibition. We thus suggest that the formation of the basal Numb crescent in dividing NB is driven by LLPS induced by the interaction between Numb and Pon.

The observation that the Numb/Pon complex in the condensed liquid phase can rapidly exchange with the corresponding proteins in the aqueous phase is consistent with the fact that Numb and Pon are in fast equilibrium between cortex crescent and cytoplasm in asymmetrically dividing *Drosophila* NBs and SOP cells[27,28]. Our observation of the Numb/Pon complex LLPS provides a mechanistic explanation to the stable existence of large concentration gradients of the proteins within the crescent and those in the cytoplasm.

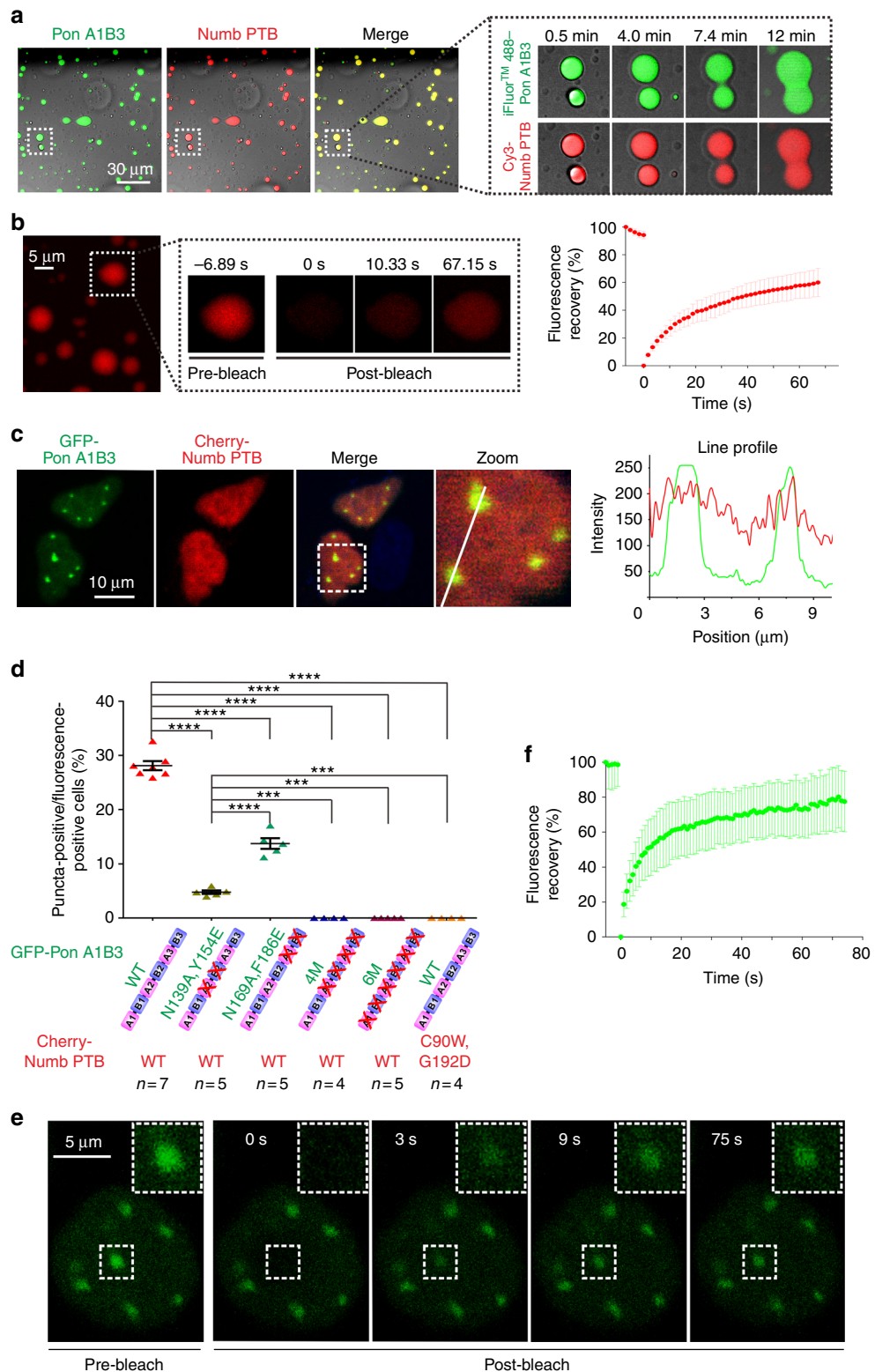

The establishment and maintenance of cell polarity in many tissues require several sets of evolutionarily conserved master polarity complexes such as the Par-3/Par-6/aPKC complex and the Lgl/Dlg/Scribble complex in the apical-basal polarity, and the Prickle/Vangl and Frizzled/Dishevelled/Diego complexes in the planer cell polarity[1,2,6–9,46]. A common hallmark of these protein complexes in polarized cells is that proteins in each of these complexes interact with each other autonomously forming locally high-concentrated patches or even puncta-like shapes[47]. Essentially, all these highly concentrated complexes are peripherally associated with the inner surface of plasma membranes and are in open contacts with aqueous cytoplasm. Proper concentration and localization of these polarity complexes are well known to be critical for cell polarity[1,2,6–8,46]. It is also well established that these polarity complexes can readily dissolve and disperse from cell cortices when cells lose polarity. All these features share very high similarity with what we have observed for the Numb/Pon complex in this work. It is tempting to speculate that some of these polarity regulatory complexes may also undergo LLPS upon complex formation and such phase transition facilitates proper localization as well as condensation of these complexes in polarized cells. This prediction will certainly need to be tested in the future.

It is increasingly recognized that the assembly of membrane-less compartments, including mitotic spindles, centrosomes, nucleoli, and various cellular bodies and RNA-enriched granules, as well as some large signal transduction machineries beneath the plasma membrane, such as the postsynaptic densities and T-cell signaling pathway, is driven by phase separation of specific components[40,43,48–59]. In a broad sense, phase transition-induced formation of these membrane-lacking organelles and signaling machineries is another kind of protein condensation, just as the formation of Numb/Pon crescent during ACD. While polarized signaling is a common phenomenon during cell polarization, e.g., the Wnt singling in axon guidance, it is likely that some of these polarized signal transduction machineries may also undergo LLPS in polarized cells. Thus, phase transition may be generally utilized to achieve polarized protein localization and signaling in cell polarity.

Thus far, ~60 PTB domain-containing proteins have been found in humans. As adapters or molecular scaffolds, PTB domain-containing proteins are involved in a wide range of signaling processes[39]. Most PTBs can recognize a consensus "NPxY"/"NxxF" type A motif (with or without phosphorylation of Tyr) in their cargos with relatively weak binding affinities (mostly Kd ~10–100 μM range, few reaches the 1 μM range)[60,61]. In a recent study, all PTB domains from 17 different proteins were capable to bind to a subset of type A motif-containing integrin cytoplasmic tails in an in vitro binding assay, whereas only a handful of these PTB proteins have been characterized to have bona fide effects on integrin-mediated cell adhesion

events[62], pointing to the fact that the isolated type A motif recognition may not be sufficient for the specific PTB-cargo interactions. Our study discovered that the synergistic interaction of a second motif (the B motif) together with the canonical A motif in proteins such as Pon and Nak greatly increases their binding affinity and selectivity toward Drosophila Numb PTB. Analysis of the available PTB structures further suggest that the newly identified B motif-binding site seems to be a common property in many PTB domains (Supplementary Fig. 5). We propose an A and B motif-mediated specific PTB/target recognition model in Fig. 3e, and such combined two site target recognition model likely provides much higher binding affinity and specificity for some PTB domains. Additionally, the existence of two binding sites on a PTB domain provide a biochemical basis for certain PTB domains such as the one in Numb to form multivalent complex assemblies with their targets. Such multivalent interaction-mediated protein complex may offer additional properties such as phase transitions in addition to enhancing the binding affinities and specificities.

## Methods

**Protein expression and purification.** Various Drosophila Pon fragments (Uniprot ID: Q9W4I7-1, Fig. 1), the Drosophila Numb PTB (Uniprot ID: P16554-1, aa 65–203), the Drosophila Nak fragments (Uniprot ID: Q9VJ30-1, Fig. 3a, b), the mouse Numb PTB (Uniprot ID: Q9QZS3-4, aa 21–157), and the rat MINT2 PTB (Uniprot ID: O35431-1, aa 365–557) were individually cloned into pGEX-6P-1 or a modified version of pET32a vector in which the thrombin-cutting site was replaced by a protease 3C-cutting site, and the S-tag was removed[63]. All the mutations used in this study were created using the standard PCR-based mutagenesis method and confirmed by DNA sequencing. Recombinant proteins were expressed in Escherichia coli BL21 (DE3) host cells at 16 °C and were purified by using a Ni²⁺-NTA agarose affinity chromatography followed by size-exclusion chromatography (SEC). SEC experiments were performed using HiLoad 26/600 superdex 75/200 pg columns on an AKTA FPLC system (GE Healthcare). Protein elution was detected by absorbance at 280 nm. Proteins were finally equilibrated in the buffer containing 50 mM Tris (pH 8.0), 100 mM NaCl, 1 mM dithiothreitol (DTT; or 1 mM β-mercaptoethanol, (β-ME)) and 1 mM EDTA. The N-terminal His-tagged fragment of Drosophila Numb PTB was cleaved by digesting the fusion protein with pre-Scission protease (50 μg protein with 1 μl protease, Sigma, GE27-0843-01) at 4 °C, and the protein was purified by another step of SEC. pPon peptide (ESCFT-NAAFSSpTPKK) was commercially synthesized and described in our earlier work[35].

**Isothermal titration calorimetry measurements.** ITC measurements were performed on an ITC200 Micro calorimeter (MicroCal) at 20 °C. All of the protein samples were dissolved in buffer A containing 50 mM Tris (pH 8.0), 1 mM EDTA, 1 mM β-ME and 100 mM NaCl. The titrations were carried out by injecting 40-μl aliquots of various Pon fragments (~0.5 mM) into indicated PTB proteins (~0.05 mM) at time intervals of 2 min. The titration data were analyzed using the program Origin 7.0 and fitted by the one-site binding model.

**Glutathione S-transferase pull-down assay.** Glutathione S-transferase (GST) or GST fusion proteins (2 nmol) was first loaded onto 40 μl GSH-Sepharose 4B slurry beads and then incubated with 6 nmol indicated proteins in 500 μl buffer A for 1 h at 4 °C. After being washed three times with the assay buffer, the above proteins captured by affinity beads were eluted by boiling, resolved by 12% SDS-

**Fig. 6** The Numb PTB/Pon A1B3 complex forms liquid droplets and undergoes constant dynamic exchanges both in vitro and in living cells. **a** The time-lapse images showing the co-localization of iFluor™ 488-Pon A1B3 and Cy3-Numb PTB in the droplets with enriched concentrations. The enlarged images at right show that small droplets can grow and merge into larger ones as time goes on. **b** FRAP analysis of Cy3-Numb PTB droplets in vitro showing the exchange kinetics of the protein in droplets with the surrounding aqueous solution (see also Supplementary Movie 4). The red curve at right represents a FRAP recovery curve by averaging signals of 23 droplets with similar sizes each after photobleaching. Time 0 refers to the time point of the photobleaching pulse. All data are represented as mean ± SD. **c** Representative images showing co-expression of GFP-Pon A1B3 and Cherry-Numb PTB in HeLa cells produce multiple bright puncta containing both fluorophores. Dashed boxes show the zoomed-in regions. **d** Numb-Pon-binding deficient Pon A1B3 mutants (N139A,Y154E, N169A,F186E, 4M, and 6M) or Numb PTB mutant (C90W,G192D) showing significantly decreased puncta formation when compared to the WT proteins (n = number of batch of cultures with >820 cells counted for each batch). Specimens' statistics are presented as mean ± SEM; ***p < 0.001 and ****p < 0.0001 using one-way ANOVA with Tukey's multiple comparison test. **e** Representative time-lapse FRAP images showing that GFP-Pon A1B3 signal within the puncta recovered within a few minutes (see also Supplementary Movie 6). **f** FARP analysis of GFP-Pon A1B3 in puncta of Hela cells. The green curve represents the averaged FRAP data of 23 puncta from 23 cells. Time 0 refers to the time point of the photobleaching pulse. All data are represented as mean ± SD

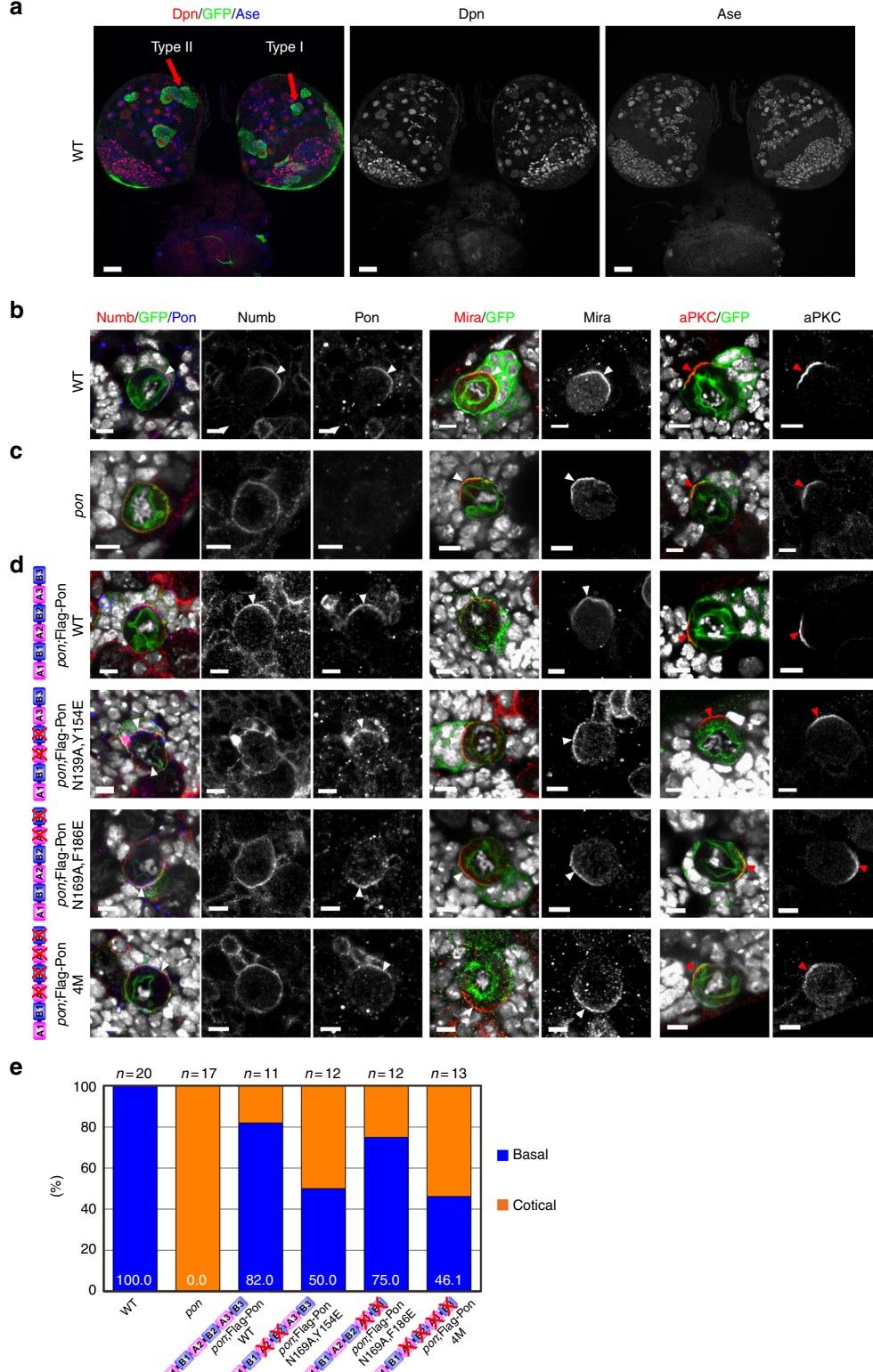

**Fig. 7** Direct interaction between Pon AB motif repeats and Numb PTB is required for Numb localization during the asymmetric divisions of *Drosophila* larval NBs. **a** Overview of fly brain showing type I and type II lineage. **b–d** NBs are marked by GFP using MARCM technique (see Methods). ToPro-3 is in white. Staining of various apical and basal proteins (red or blue) and GFP (green) in larval NBs derived from WT (**b**), *pon* mutant (**c**), and *pon* mutant expressing a Flag-Pon WT or mutant variants (**d**). **b** Numb and Pon are basally localized in wild-type NBs. **c** Pon is not detected in *pon* mutant NBs, whereas Numb is uniformly localized on cortex as well as some cytoplasm in *pon* mutant NBs. Mira and aPKC are normally localized in *pon* mutant NBs. **d** In *pon* mutant NBs rescued with Flag-Pon WT, majority (82%) of NBs exhibit basal localization of Pon and Numb. Interestingly, in *pon* mutant NBs rescued with Flag-Pon N139A,Y154E, Flag-Pon N169A,F186E, or Flag-Pon 4M, majority of these NBs showing broad cortical as well as some cytoplasmic localization of Numb, although all the Flag-Pon variants are basally localized. Mira and aPKC are normally localized in these NBs. White arrowheads label basal cortex, whereas red arrowheads indicate apical cortex. Scale bars, 25 μm for the whole brain in **a** and 5 μm for rest NB images. **e** Statistical data of Numb localization for **b–d**

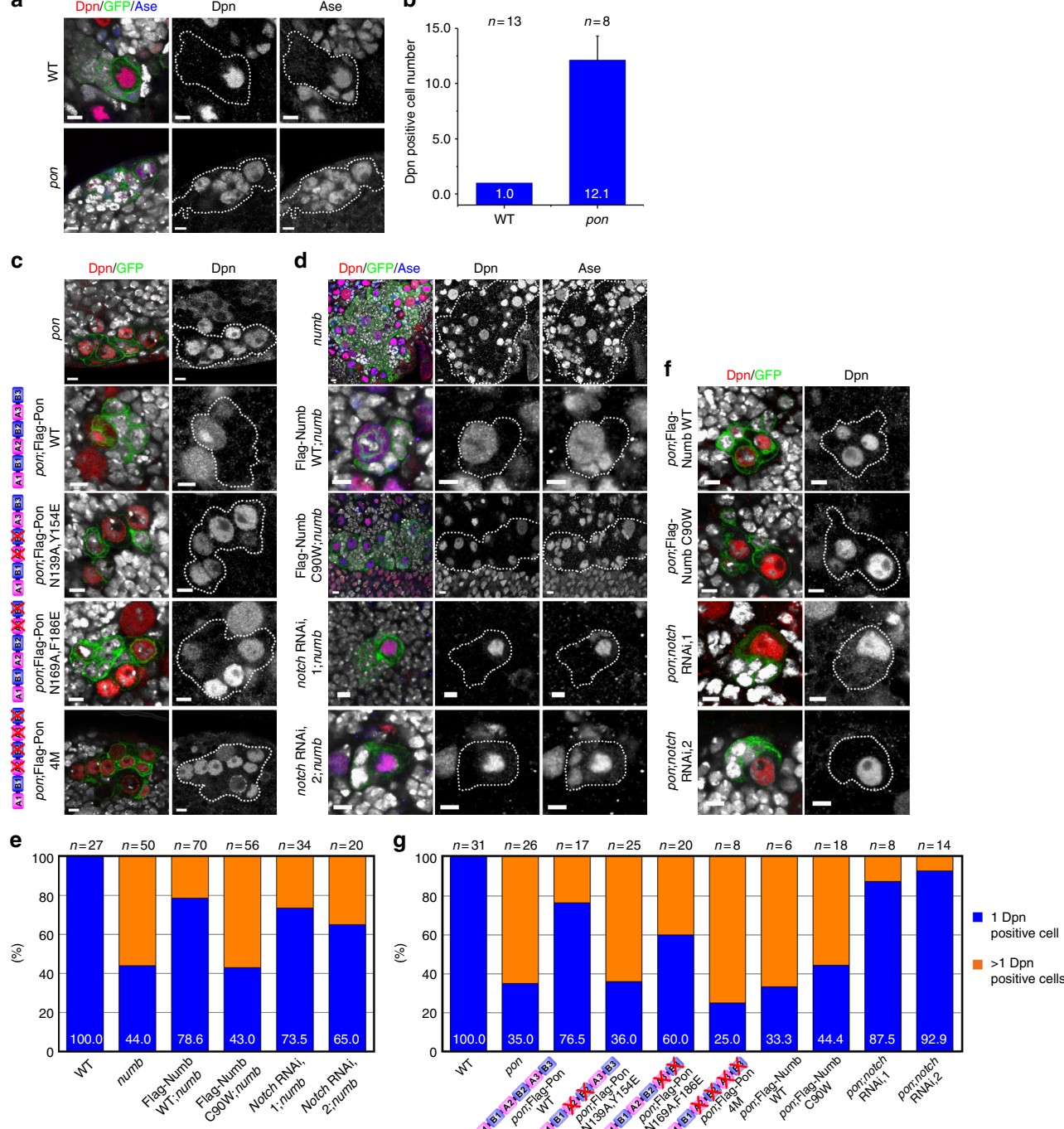

**Fig. 8** Pon-Numb interaction is required for Numb function in inhibiting Notch signaling. **a–f** NBs are marked by GFP using MARCM technique. Dpn is in red, GFP in green, and Ase in blue. **a** *pon* mutant clone showing more Dpn- and Ase-positive NB-like cells than the wild-type counterpart. **b** Quantification of results in **a** showing that type I pon mutant NB clone (12.13 + 2.03, *n* = 8) in central brain contains more Dpn-positive cells, compared to WT clone (1.00 + 0.0, *n* = 13, *p* < 0.001, mean + SEM). Data are evaluated with Student's *t*-test. **c** Rescue of *pon* mutant with Flag-Pon WT or variants. The over-proliferative phenotype of *pon* mutant clone could be rescued with *Flag-Pon WT* transgene, and partially rescued with *Flag-Pon N169A,F186E* transgene, but could not be rescued with *Flag-Pon N139A,Y154E* or *Flag-Pon 4M* transgene. **d** *numb* mutant NB lineage harboring multiple Dpn- and Ase-positive cells that is reverted by expression of Flag-Numb WT but cannot be rescued by the Pon-binding deficient Flag-Numb C90W variant. Introduction of *notch* RNAi (bl27988 for RNAi-1 and bl31180 for RNAi-2) could rescue the over-proliferative phenotype of *numb* mutant. **e** Statistical data for **d**. **f** The *pon* mutant phenotype could only be rescued by introducing *notch* RNAi, but not by expressing a *Flag-Numb WT* or *Flag-Numb C90W* transgene. Scale bars, 5 μm. **g** Statistical data for **c** and **f**

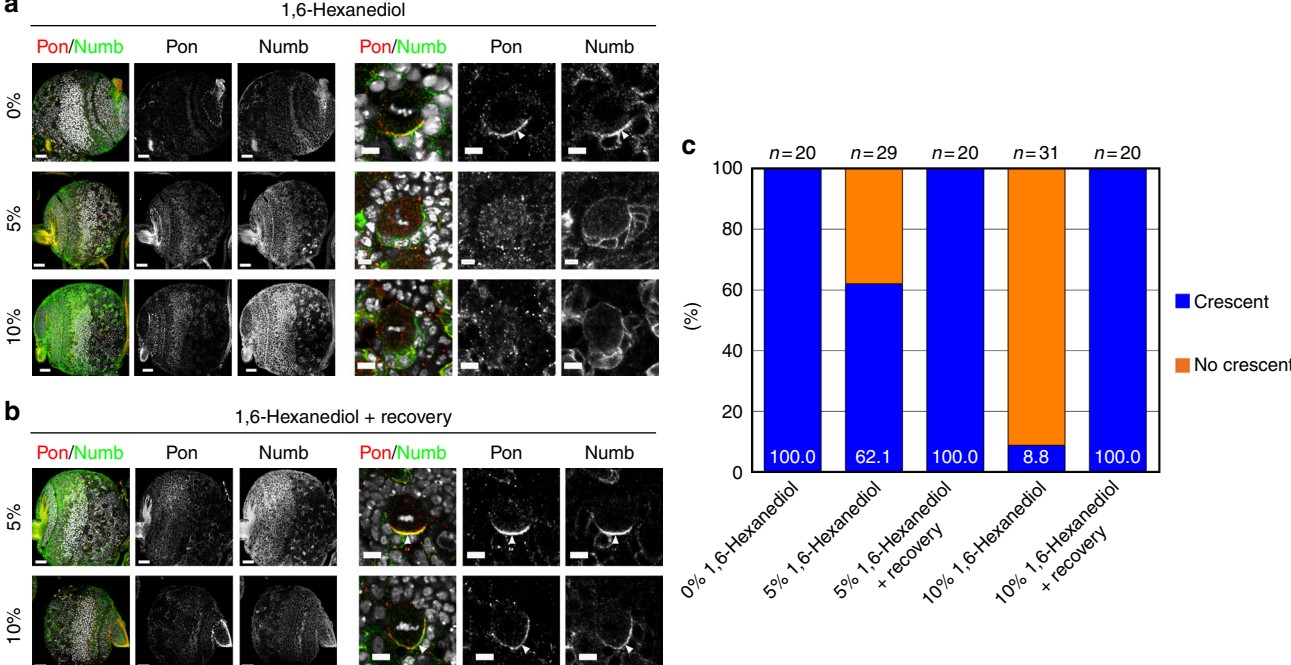

**Fig. 9** The basal localization of Numb and Pon is disturbed by 1,6-hexanediol in dividing NBs of larval brains. **a** In wild-type NBs treated with 5% 1,6-hexanediol, some (37.9%) NBs exhibit cytoplasmic localization of Pon and cortical localization Numb. When treated with 10% 1,6-hexanediol, virtually all NBs show cytoplasmic Pon and cortical Numb. **b** After removal of 1,6-hexanediol, the basal distribution of Numb/Pon is restored in NBs pre-treated with hexanediol. ToPro-3 in white. White arrowheads point to basal cortex. Scale bars, 25 µm for the whole brain and 5 µm for NB images. **c** Statistical data for **a** and **b**

polyacrylamide gel electrophoresis (SDS-PAGE), and detected by Coomassie blue staining. Uncropped gels are shown in Supplementary Fig. 10.

**Crystallography.** Freshly purified *Drosophila* Numb PTB was concentrated to 6–15 mg/ml before adding Pon peptides (10 mM stock solutions in 50 mM Tris (pH 8.0), 100 mM NaCl, 1 mM DTT, and 1 mM EDTA) with a molar ratio of 1:3. Crystals of the PTB/Pon A2B2 and PTB/Pon B1A2 complexes were grown by the hanging drop vapor diffusion method at 16 °C in a reservoir solution containing 0.1 mM sodium formate (pH 7.0), 12% w/v polyethylene glycol 3350 or 0.1 mM Bis-Tris (pH 4.3), 0.2 mM magnesium chloride hexahydrate, and 25% w/v polyethylene glycol 3350, respectively. The crystals were soaked in crystallization solution containing 20% or 6% glycerol, respectively, for cryoprotection. The diffraction data of the crystals of PTB/Pon A2B2 and PTB/Pon B1A2 complexes were collected at the beamlines BL18U1 and BL17U1 at Shanghai Synchrotron Radiation Facility in China and National Center for Protein Sciences Shanghai at wavelength of 0.9754 and 0.9792 Å, respectively. The data were processed and scaled using HKL3000 and HKL2000[64]. The phase problem of the PTB/Pon A2B2 and PTB/Pon B1A2 complexes were solved by molecular replacement using the program PHASER[65] with the human Numb-like protein PTB structure (PDB ID: 3F0W) as the search model before was adjusted by COOT[66]. For the PTB/Pon A2B2 complex, only one copy of PTB and Pon A2B2 was found in an asymmetric unit, whereas for the PTB/Pon B1A2 complex, two copies of PTB and Pon B1A2 exist in an asymmetric unit. The initial model was further rebuilt, adjusted manually with COOT, and refined by the phenix.refine program of PHENIX[67]. The final model of the PTB/Pon A2B2 and PTB/Pon B1A2 complexes, respectively, have 98.7% and 100% of the residues in the favored region of the Ramachandran plot with no outliers. The statistics of the data collection and final refinement statistics are summarized in Table 1.

**Coimmunoprecipitations and immunoblotting.** Human HEK293T cells (from American Type Culture Collection (ATCC)) were grown in Dulbecco's modified Eagle medium (Hyclone) containing 10% fetal bovine serum (FBS; Hyclone). Cells were transiently co-transfected with indicated full-length Pon and *Drosophila* Numb or various mutants using polyethylenimine transfection reagent (Polysciences). Cells were harvested 36 h post transfection and lysed in a buffer containing 50 mM Tris (pH 7.4), 150 mM sodium chloride, 1% Nonidet P-40, 10 mM sodium fluoride, 1 mM sodium metavanadate, 1 mM phenylmethylsulfonyl fluoride and protease inhibitors. Each lysate was incubated with anti-Flag M2 affinity gel (Sigma) overnight. After extensive wash with the lysis buffer, the above proteins captured by affinity beads were used for immunoblotting.

The captured proteins were boiled in SDS-PAGE loading buffer and subjected to SDS-PAGE. The proteins were transferred to a 0.45 µM polyvinylidene difluoride (PVDF) membrane (Millipore), which was blocked using 3% bovine serum albumin in TBST (20 mM Tris-HCl (pH 7.4), 137 mM NaCl, and 0.1% Tween-20) buffer at room temperature for 1 h; this was followed by incubation with the following antibodies: anti-Flag (ABclonal, AE005); or anti-HA (ABclonal, AE008), at a 1/2000 dilution at 4 °C overnight. Membranes were washed three times with TBST buffer, incubated with horseradish peroxidase-conjugated goat anti-mouse antibody (ABclonal, AS003), and visualized on a LAS3000 Chemiluminescence Imaging System.

**In vitro phase transition assay.** Various *Drosophila* Pon fragments and Numb PTB (WT or mutants) were purified in buffer A pre-cleared via high-speed centrifugations for 10 min. In the assay, the two proteins were mixed at indicated molar ratio at final concentrations spanning 5–100 µM. Formations of phase transition were assayed either directly by imaging-based methods or by sedimentation-based methods.

For imaging, mixtures were injected into a self-made flow chamber comprised of a glass slide sandwiched by a coverslip with one layer of double-sided tape as a spacer for DIC (Zeiss) or fluorescent imaging (Leica SP5 or SP8). For the sedimentation assay, samples were subjected to centrifugation at 14 000 r/min for 10 min at 4 °C. Supernatant was isolated from pellet into a clean tube immediately after centrifugation. The pellet fraction was washed once with protein buffer and thoroughly re-suspended with the same buffer to the equal volume as supernatant fraction. Proteins from both fractions were detected by 12% SDS-PAGE with Coomassie blue staining. Band intensities were quantified using Typhoon FLA 9500 (GE HealthCare).

For fluorescence assay, Pon A1B3 and Numb PTB were purified in buffer containing 100 mM NaHCO₃ (pH 8.3), 100 mM NaCl, and 1 mM DTT. iFluor™ 488 NHS ester (Tianjin Biolite Biotech) and Cy3 NHS ester (AAT Bioquest) were incubated with Pon A1B3 or Numb PTB, respectively, at room temperature for 1 h (fluorophore to protein molar ratio was 3:1). Reaction was quenched by 200 mM Tris (pH 8.0). Chemical-labeled proteins were further purified into buffer A by Hitrap desalting column. In flow chamber at room temperature, 1:3 mixture of Pon A1B3 (100 µM, with iFluor™ 488-labeled Pon A1B3 mixed with 10 molar ratios of unlabeled molecules) and Numb PTB (300 µM, with Cy3-labeled molecules mixed with 600 molar ratios of unlabeled molecules) was observed with a Leica SP5 confocal microscope.

**HeLa cell imaging and data analysis.** HeLa cells (from ATCC) were seeded on 25-mm poly-L-lysine-coated coverslips. For each well in a 12-well plate, various GFP-Pon A1B3 (0.2 µg) and mCherry-Numb PTB (0.8 µg) plasmids were

individually or co-transfected into HeLa cells using ViaFect™ transfection Reagent (Promega). Cells were fixed by 4% paraformaldehyde and mounted on glass slides for imaging using a Zeiss LSM 880 confocal microscope by a ×40 oil-immersion lens with 4′,6-diamidino-2-phenylindole staining. Confocal images were processed with ImageJ. For puncta-counting assay, data were collected from 4–7 independent batches of cultures as indicated in the figure. In each batch, at least >820 fluorescence-positive cells were counted for each group of experiments. A cell with more than two obvious fluorescent puncta was counted as a puncta-positive cell. Experiments were conducted in a blinded fashion.

**Fluorescence recovery after photobleaching assay**. The in vitro FRAP analysis of Cy3-Numb PTB droplets was carried out in a 1:3 mixture of Pon A1B3 (100 μM) and Cy3-Numb PTB (300 μM) at room temperature. The Cy3 signal was bleached using a 561-nm laser beam with a Leica SP8 confocal microscope.

HeLa cells were cultured in glass-bottom dishes (MatTek) and co-transfected as described above. FRAP assay was performed on a Zeiss LSM 880 confocal microscope supported with a Chamlide TC temperature, humidity, and $CO_2$ chamber. Puncta with diameters around 1.0 μm were assayed. GFP signal was bleached using a 488-nm laser beam. The fluorescence intensity difference between pre-bleaching and at time 0 (the time point right after photobleaching pulse) was normalized to 100%. The experimental control is to quantify fluorescence intensities of similar puncta/cytoplasm regions without photobleaching.

**Analytical ultracentrifugation analysis**. Sedimentation velocity (SV) experiments were performed in a Beckman Coulter XL-I analytical ultracentrifuge using double sector centerpieces and sapphirine windows. SV experiments were conducted at 42,000 rpm and 4 °C using interference detection. The SV data was analyzed using the SEDFIT program[68].

***Drosophila* S2 cell culture**. Pon (Flag-Pon WT, Flag-Pon N139A,Y154E, Flag-Pon N169A,F186E, and Flag-Pon 4M) and Numb (Flag-Numb WT and Flag-Numb C90W) variants were subcloned into UASt.attB vector (a gift from Konrad Basler).

*Drosophila* S2 cells (from *Drosophila* Genomics Resource Center) were grown at 25 °C in Schneider's medium (Invitrogen) supplemented with 10% FBS. All transfections were performed using Effectene Reagent (Qiagen) according to the manufacturer's instructions. Briefly, S2 cells were co-transfected with 0.5 μg plasmids of interest (Pon or Numb variants) together with *act-gal4* plasmid. Cells were harvested at 48 h later and lysed in Nonidet P-40 lysis buffer containing 50 mM Tris, pH 8.0, 250 mM NaCl, 0.5% Nonidet P-40, 0.2 mM EDTA, protease inhibitor cocktail (complete, Roche), and phosphatase inhibitor. The lysate was collected and cleared by centrifugation at 13,000 rpm for 5 min at 4 °C. The samples were separated by 10% polyacrylamide SDS-PAGE gels followed by transferring to PVDF membranes (Millipore). Mouse anti-Flag antibody (Sigma, F1804, 1/2000), rabbit anti-Numb antibody (a gift from Yuh-Nung Jan, 1/1000), and rabbit anti-Pon (generated in our lab, 1/2000) were diluted in TBST with 5% non-fat dry milk.

**Fly genetics**. Information about the fly stains used in this study was described in the text or FlyBase (www.flybase.org). Stocks (unless stated below) were obtained from Blooming Stock Center and crosses were maintained at 25 °C on standard medium. Stocks used were *FRT19A*, *FRT42B*, *elav-gal4*, *insc-gal4*, *Tub-gal80*, *hs-flp*, *UAS-CD8::GFP*, *numb[796]*, *notch RNAi-1 (bl27988)*, *notch RNAi-2 (bl31180)*, and *pon[p26]* (a gift from Yuh-Nung Jan).

Pon (Flag-Pon WT, Flag-Pon N139A,Y154E, Flag-Pon N169A,F186E, and Flag-Pon 4M) and Numb (Flag-Numb WT and Flag-Numb C90W) variants were subcloned into UASt.attB vector and transgenic lines were generated by BestGene, Inc. (ChinoHills,CA) using *attP* landing site on II chromosome (Best Gene line 9723).

To address these Pon and Numb variant localization and function in *pon* or *numb* mutant background, these transgenes were subsequently crossed into *FRT19A*, *FRT19A.pon[p26]* or *FRT42B.numb[796]* background.

Mosaic analysis with a repressible cell marker (MARCM) technique was used to positively mark mutant clones with a GFP signal according to published protocol[69]. In brief, embryos were collected over a period of 6 h, and larvae (24 h after larval hatching, ALH) was subjected to 1-h heat-shocked treatment at 37 °C, and larvae with desired genotypes were dissected and examined.

**Immunohistochemistry and imaging**. Larvae of desired genotype were dissected at 96 h ALH and brains were fixed for 15 min in 3.7% formaldehyde in PBS with 0.1% Triton-X, and later processed for immunochemistry analysis. The following antibodies were used: mouse anti-flag (Sigma), 1/2000; rabbit anti-Mira[70] (generated in our lab), 1/1000; guinea-pig anti-Dpn (generated in our lab), 1/1000; rabbit anti-Pon (generated in our lab), 1/2000; rabbit anti-Numb, 1/1000; rabbit anti-aPKCζ C20 (Santa Cruz Biotechnologies, SG-216-G), 1/1000; chicken anti-GFP (Abcam, ab13970), 1/5000; anti-NICD (DSHB, C17.9C6), 1/50; anti-NECD (DSHB, C458.2H), 1/50; and anti-Ase (generated in our lab), 1/1000. Secondary antibodies were conjugated to Alexa Fluor 488, Alexa Fluor 555, or Alexa Fluor 633 (Molecular Probes), and used at 1/500, 1/1000, and 1/250, respectively. TO-PRO-3 (Invitrogen) was used at 1/5000 for DNA staining and samples were mounted in

Vectashield (Vector Laboratories). Images were obtained using Leica SP8 upright microscope and processed in Adobe Photoshop CS6 and Adobe Illustrator CS6.

**Effect of 1,6-hexanediol on Numb/Pon assemblies**. To analyse the effect of 1,6-hexanediol on Numb/Pon assemblies in vivo, larvae of $w^{1118}$ at 96 h ALH were dissected in Shields and Sang M3 insect medium (Sigma) and larval brains were transferred to Shields and Sang M3 insect medium containing 0, 5, or 10% 1,6-hexanediol, and incubated for 2 min. Then the treated brains were fixed immediately for 15 min in 3.7% formaldehyde in PBS with 0.1% Triton-X, and then processed for immunochemistry analysis. For recovery experiment, the hexanediol-treated brains were washed with M3 insect medium for several times and incubated in Shields and Sang M3 insect medium for 20 min before fixation. The following antibodies were used: rabbit anti-Pon (generated in our lab), 1/2000; and rabbit anti-Numb (generated in our lab), 1/1000. Pon and Numb localization were analysed under Leica Confocal microscopy SP8 system. Only mitotic NBs were analysed.

**Quantification and statistical analysis**. Statistical parameters including the definitions and exact values of *n* (e.g., number of experiments, number of cells, etc.) are reported in the figures and corresponding figure legends. Data of HeLa cell culture were expressed as mean ± SEM; ***$p < 0.001$ and ****$p < 0.0001$ using one-way analysis of variance (ANOVA) with Tukey's multiple comparison test. Data of in vitro phase transition sedimentation assay and FRAP assay were expressed as mean ± SD. Data are judged to be statistically significant when $p < 0.05$ by one-way ANOVA with Tukey's multiple comparison test. None of the data were removed from our statistical analysis as outliers. For protein localization in NBs of various genetic backgrounds, only mitotic NBs were imaged and counted. For counting of Dpn-/Ase-positive cells in MARCM clones, z-series images were acquired and cells were counted. All statistical data was conducted in GraphPad Prism 5. All experiments related to cell cultures and imaging studies were performed in blinded fashion.

**Data availability**. The authors declare that all data supporting the findings of this study are available within the article and its Supplementary Information files or from the corresponding author upon reasonable request. Coordinates of the crystal structures of Numb PTB/Pon A2B2 and Numb PTB/Pon B1A2 have been deposited in the Protein Databank under the accession code 5YI8 and 5YI7, respectively.

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

## Acknowledgements

We thank the staff of beamlines BL17U1 and BL18U1 at SSRF and NCPSS for data collection; Dr. Jiayi Zhang for helping with DIC imaging; and Konrad Basler, Yuh-Nung Jan, Bingwei Lu, the Bloomington Stock Center for antibodies, vector, and stocks. This work was supported by grants from the Ministry of Science and Technology of the People's Republic of China (2014CB910201 and 2014CB910204), the National Natural Science Foundation of China (31422015, 31670730, and 31270778), and the Shanghai Municipal Education Commission (14SG06) to W.W., Temasek Life Sciences Laboratory

and Singapore Millennium Foundation to Y.C., and grants from RGC of Hong Kong (AoE-M09-12) to M.Zhang.

## Author contributions

Z.S., Y.T., Y.Y., M.Zhang, Y.C., and W.W. conceived the research and analyzed data. Z.S., Y.T., and Z.L. performed the biochemical experiments. Z.S. solved the crystal structure, and carried out Co-IP and phase transition experiments. M.Zeng and M.Zhang helped with the phase transition assay. Y.Y. and Y.C. undertook analysis of transgenic flies. H.X. and J.L. performed the analytical ultracentrifugation measurements. W.W. drafted the manuscript and all authors commented on it. W.W. coordinated the research.

## Additional information

**Competing interests:** The authors declare no competing financial interests.

