## [Peer Review File · Nature Communications]

Reviewers' comments:

Reviewer #1 (Remarks to the Author):

Shan et al set out to study the molecular underpinnings of asymmetric cell divisions using a combination of structural biology, biochemical, cellular and in vivo fly assays. The asymmetric distribution of the membrane associated Numb protein in *Drosophila* neuroblast mother cells leads to the acquisition of different cell fates in daughter cells after cell division. Pon (Partner of Numb) has been proposed to contribute to the asymmetric localization of Numb at the basal cortex. How the Numb protein is selectively distributed is therefore a key molecular event to understand the basis for asymmetric cell division. A key achievement of this study are two new 'dimer-of-dimer' structures consisting of two phosphotyrosine binding (PTB) domains of Numb and a 16-meric Pon peptide providing the molecular basis for the interaction. Apart from a previously described A-site mediating Numb/Pon interactions a new B-site has been identified facilitating multivalent interaction and Pon-mediated tethering of Numbs. Interestingly, the relative orientations of Numb PTBs (head-to-head vs tail-to-tail) depend on the identity of the A and B peptides (Pon consist of an A1B1A2B2A3B3 sequence of these peptides). The structural data informed the design of specific mutations that interfere with complex formation. The mutant proteins enabled a series of assays to study the relevance of these interactions for liquid-liquid phase separation and subcellular localization in cell culture and fly brains. Key claims of the work are that Pon utilizes the A and B sites on the Numb PTB to aggregate the complexes leading to liquid-liquid phase separation which is the underlying mechanism for the localization of Numb at the basal crescent and asymmetric cell division. The manuscript is well written, technically sound and touches on several interesting aspects. It includes a broad range of techniques and elegantly combines structural analysis with functional assays. Nevertheless, I believe the clarity of the presentation can be improved and the main claims could be strengthened by better linking structural and functional data. To me it remains somewhat unclear which configurations of Numb/Pon complexes mediate the phase separation. In other words, it would be of interest to understand how the 'dimers-of-dimers' are connected within the expected mesh of phase separated complexes. Is the higher order complex related in some form to Figure S3d? If the authors could come up with the model backed by the body of data of how the full-length PON (A1-B3) bridges adjacent Numb molecules that would strengthen the study.

Specific points

- 1) The introduction should make clear what Numb and Pon structures have been reported previously and how the present structures go beyond these models. Several Numb structures are available in the pdb and a Pon/Plk1 complex structure has been reported as well by the same authors (5j19).
- 2) Figure 6b: It could be useful to show the IHC WT controls for Mira and aPKC as a reference.
- 3) I suggest adding the cartoons used in Figures 2d/e/f to Figures 4-7 as well (possibly as legend to, for example, Figure 4c). I found myself flipping back and forth to understand what mutants or truncated Pon versions are being investigated.
- 4) Along these lines it maybe helpful to provide a summarizing supplementary table that shows which constructs/mutants were studied and how they behaved in the various assays performed. This may also help to come up with a model for the higher order Numb/Pon complex expected to induce phase separation.
- 5) The A1A3 construct (that caused phase separation) has not been used in ITC and (Figure 1bc) and SV experiments (Figure 3c). Please explain why.

6) It seems the color coding for the peptides changed in Figures 2a-c. For example, The color of Numb in Figure 2b resembles PonA2 in Figure 2c which is confusing. I suggest to adjust coloring and to also consider consistency with the color of the cartoons used in Figure 2d-f for clarity.

7) I don't understand how Figure S4 supports the omnipresence of the B motif. The molecular interfaces seem very different. Perhaps authors could highlight the B-motif (NxxF/NPxY) in all the structures and describe what makes these structural interfaces similar?

8) Methods for protein production and crystallography should provide more details:

- 1) Uniprot IDs should be given for the protein constructs used.
- 2) What buffers were used to purify proteins (lysis, IMAC, SEC). In which buffer and at which concentration was the PON peptide dissolved in?
- 3) Provide source of precision enzyme and protease-substrate ratio
- 4) Which SEC column was used?
- 5) What molar ratio of PTB to PON was used to set up crystallization drops?
- 6) The structure was reported to be solved by MR. How was the search model prepared and how many copies were searched for in the M.U.? What software was used for MR?
- 7) How many 'dimer-of-dimers' are in the AU?
- 8) What is FMT in Table S1?
- 9) What is the range of the highest resolution shell in Table S1?
- 10) Pdb accession numbers should be provided and coordinates as well structure factors submitted to the pdb.

Reviewer #2 (Remarks to the Author):

In this manuscript the authors presented biochemical and genetic evidences supporting the model that multivalent interaction between repeat PTB-binding motifs in Pon and the PTB domain of Numb promotes phase-separation of the PON-Numb localization complex, and that this process is instrumental in directing the asymmetric localization of Numb and the asymmetric cell fate determination in the *Drosophila* NB lineages. Liquid-liquid phase separation (LLPS) is increasingly being recognized as a biophysical process involved in asymmetric cell division, for example the segregation of P granules in the early division of *C. elegans* embryos. The implication of LLPS in the asymmetric segregation of Numb is thus a timely topic and of general interest to people working in the field of asymmetric cell division. Overall the studies are carefully performed and the results are of high quality. There are a few comments the authors may want to consider to make this a stronger manuscript suitable for *Nat Commun*.

1. Overall, the *in vivo* evidence of LLPS involvement in Numb asymmetric localization of weak. Cultured HeLa cells and colocalization of Numb and Pon in the nucleus are used as the readout. How about the Numb-Pon interaction at the cell cortex of dividing NBs, which is the more relevant physiological setting?

2. Aliphatic alcohols, for example 1, 6 hexanediol has been used to disturb labial, phase-separating assemblies. Can the PON-Numb assemblies *in vitro* and *in vivo* be melted by Aliphatic alcohols?

3. Is it possible to use exogenous LLPS forming motifs from other proteins to replace the PON-Numb interacting motifs and rescue the Numb localization defect in Pon mutant. This would be the ultimate test of the role of LLPS in Numb localization.

4. It is surprising that mouse Numb PTB domain does not interact with the PTB-binding motifs in Pon, although mammalian Numb has been shown to be asymmetrically localized in fly NSCs and rescue Numb mutant phenotypes. The authors should at least discuss this point.

Reviewer #3 (Remarks to the Author):

Comments on "Phase transition-mediated Numb/Pon complex basal condensation during neuroblast asymmetric division" by Wen et. al

This manuscript comprehensively investigated the interaction between the PTB domain of *Drosophila* Numb and one of its binding partner, Pon, through crystallographic, other biophysical and biochemical studies. They fully characterize the two binding surfaces of PTB: the canonical surface associating with one of three type A motifs and the other, a non-canonical surface, binding one of three type B motifs of Pon's N-terminus. Interestingly such a bipartite binding mode on PTB not only ensures high specificity/affinity interaction between Numb and Pon but also, together with repeating linear motifs of Pon, enables multivalency-driven liquid-liquid phase transition (LLPT) by the two in vitro and in Hela cell. Furthermore, the authors show that perturbations (i.e. mutations of Numb or Pon) to LLPT also interfere with some essential functions of Numb and/or Pon, such as the control of Numb polarized location by Pon and the function of these two towards inhibiting Notch activities in fly.

If the following concerns are adequately addressed, I find this manuscript of the significance and general interest to be published in Nature Communication.

A major point:

Liquid-liquid phase transition of biomacromolecules is quickly emerging as a field of research in biology. This field was largely initiated by the combination of two landmark publications: one in Science in 2009 and the other in Nature in 2012. The latter paper is especially relevant here because it has explicitly established the multivalency-driven LLPT concept in biology. The manuscript cites a few recent publications while missing the two landmark papers, which are pretty recent as well.

Minor points:

1. Figure 1b shows that the binding affinities between Numb PTB and Pon's di-motif constructs are consistently higher than those between PTB and Pon's tri-motif constructs. It hardly makes thermodynamic sense. It is likely due to a wrong model used for ITC data fitting. Therefore, a discussion of this apparent inconsistency is needed to avoid confusion. Alternatively, the latter set of ITC data shall be left out, which doesn't affect any of the major conclusions of the paper.

2. The observed phase transition is in three-dimension in vitro and in Hela cell while the crescent shaped Numb/Pon complex is more or less in two-dimension in fly. A possible mechanism leading to dimension reduction has to be discussed. For example, a membrane associating domain within or a lipid modification on Numb or Pon or a third partner of the Numb/Pon complex is responsible for the apparent membrane attachment/localization.

(Our responses to the reviewers' comments are shown in italics and highlighted in blue):

Reviewer #1:

Shan et al set out to study the molecular underpinnings of asymmetric cell divisions using a combination of structural biology, biochemical, cellular and in vivo fly assays. The asymmetric distribution of the membrane associated Numb protein in *Drosophila* neuroblast mother cells leads to the acquisition of different cell fates in daughter cells after cell division. Pon (Partner of Numb) has been proposed to contribute to the asymmetric localization of Numb at the basal cortex. How the Numb protein is selectively distributed is therefore a key molecular event to understand the basis for asymmetric cell division. A key achievement of this study are two new 'dimer-of-dimer' structures consisting of two phosphotyrosine binding (PTB) domains of Numb and a 16-meric Pon peptide providing the molecular basis for the interaction. Apart from a previously described A-site mediating Numb/Pon interactions a new B-site has been identified facilitating multivalent interaction and Pon-mediated tethering of Numbs. Interestingly, the relative orientations of Numb PTBs (head-to-head vs tail-to-tail) depend on the identity of the A and B peptides (Pon consist of an A1B1A2B2A3B3 sequence of these peptides). The structural data informed the design of specific mutations that interfere with complex formation. The mutant proteins enabled a series of assays to study the relevance of these interactions for liquid-liquid phase separation and subcellular localization in cell culture and fly brains. Key claims of the work are that Pon utilizes the A and B sites on the Numb PTB to aggregate the complexes leading to liquid-liquid phase separation which is the underlying mechanism for the localization of Numb at the basal crescent and asymmetric cell division. The manuscript is well written, technically sound and touches on several interesting aspects. It includes a broad range of techniques and elegantly combines structural analysis with functional assays. Nevertheless, I believe the clarity of the presentation can be improved and the main claims could be strengthened by better linking structural and functional data. To me it remains somewhat unclear which configurations of Numb/Pon complexes mediate the phase separation. In other words, it would be of interest to understand how the 'dimers-of-dimers' are connected within the expected mesh of phase separated complexes. Is the higher order complex related in some form to Figure S3d? If the authors could come up with the model backed by the body of data of how the full-length PON (A1-B3) bridges adjacent Numb molecules that would strengthen the study.

We agree with the reviewer that the higher order Numb/Pon complex should be related in some form to Supplementary Fig. 3d. However, due to the lack of the structural information of the Numb PTB/Pon A1B1 and Numb PTB/Pon A3B3 complexes, we could not build the higher order Numb/Pon A1B3 assemblies accurately, as all three pairs of AB motifs contribute to efficient phase separation. Even though, it is gradually accepted that multivalent interaction could be an important driving force to form mesh of phase separated complexes (Li, Nature, 2012; Su, Science, 2016; Zeng, Cell, 2016).

Specific points

1) The introduction should make clear what Numb and Pon structures have been reported previously and how the present structures go beyond these models. Several Numb structures are available in the pdb and a Pon/Plk1 complex structure has been reported as well by the same

authors (5j19).

We have modified the text following this reviewer's comments (see page 4).

2) Figure 6b: It could be useful to show the IHC WT controls for Mira and aPKC as a reference.

We have added these controls following the reviewer's suggestion.

3) I suggest adding the cartoons used in Figures 2d/e/f to Figures 4-7 as well (possibly as legend to, for example, Figure 4c). I found myself flipping back and forth to understand what mutants or truncated Pon versions are being investigated.

We have modified the figures following the reviewer's suggestion.

4) Along these lines it maybe helpful to provide a summarizing supplementary table that shows which constructs/mutants were studied and how they behaved in the various assays performed. This may also help to come up with a model for the higher order Numb/Pon complex expected to induce phase separation.

We thank the reviewer for the suggestion and have added the information in Supplementary Table 1.

5) The A1A3 construct (that caused phase separation) has not been used in ITC and (Figure 1bc) and SV experiments (Figure 3c). Please explain why.

The A1B3 construct is very unstable, easily to be degraded (Fig. 1d), and prone to aggregate. Thus, it is not suitable for ITC and SV analysis. We had tried to record the SV profile of Pon A1B3 with 1:3 molar ratio of Numb PTB (Fig. 1). As the minimal protein (complex) concentration for the SV experiment is 0.5 mg/mL (10 μ M A1B3 and 30 μ M PTB), a portion of the complex proteins went phase separation (Fig. 4d) and the supernatant displayed heterogeneous large molecular weight assemblies. However, due to the poor protein behavior (most complex proteins precipitated or aggregated, see the abnormally low c value in the y-axis, Fig. 1), the molecular weight of these large assemblies could not be accurately calculated.

Fig. 1. SV experiment of Trx-Pon A1B3 (10 uM)/Numb PTB (30 uM) complex.

6) It seems the color coding for the peptides changed in Figures 2a-c. For example, The color of Numb in Figure 2b resembles PonA2 in Figure 2c which is confusing. I suggest to adjust coloring and to also consider consistency with the color of the cartoons used in Figure 2d-f for clarity.

We have modified the figure following the reviewer's suggestion.

7) I don't understand how Figure S4 supports the omnipresence of the B motif. The molecular interfaces seem very different. Perhaps authors could highlight the B-motif (NxxF/NPxY) in all the structures and describe what makes these structural interfaces similar?

I think there may be some misunderstanding. "NxxF/NPxY" is the A motif. Supplementary Fig. 4 showed that most PTBs (with known structure) contain a hydrophobic pocket, which accommodates the conserved F/Y residue in the B motif (Fig. 1a and Supplementary Fig. 4). Thus we proposed that the bidentate target binding mode (by combining the A and B sites) may be a common feature for PTB domains. It is noted that though this hydrophobic pocket is omnipresent, the surrounding features are quite different, e.g. the acidic residue (E104 in Drosophila Numb PTB) for interacting with the conserved R residue in the B motif (Fig. 1a and 2c) is not conserved in most PTBs (Supplementary Fig. 4). Thus we proposed that the differences among the B motif binding sites from distinct PTBs provide the specificity.

8) Methods for protein production and crystallography should provide more details:

- 1) Uniprot IDs should be given for the protein constructs used.
- 2) What buffers were used to purify proteins (lysis, IMAC, SEC). In which buffer and at which concentration was the PON peptide dissolved in?
- 3) Provide source of precision enzyme and protease-substrate ratio
- 4) Which SEC column was used?
- 5) What molar ratio of PTB to PON was used to set up crystallization drops?
- 6) The structure was reported to be solved by MR. How was the search model prepared and how many copies were searched for in the M.U.? What software was used for MR?
- 7) How many 'dimer-of-dimers' are in the AU?
- 8) What is FMT in Table S1?
- 9) What is the range of the highest resolution shell in Table S1?
- 10) Pdb accession numbers should be provided and coordinates as well structure factors submitted to the pdb.

We have now added the information in the revised manuscript.

Reviewer #2:

In this manuscript the authors presented biochemical and genetic evidences supporting the model that multivalent interaction between repeat PTB-binding motifs in Pon and the PTB domain of Numb promotes phase-separation of the PON-Numb localization complex, and that this process is instrumental in directing the asymmetric localization of Numb and the asymmetric cell fate determination in the *Drosophila* NB lineages. Liquid-liquid phase separation (LLPS) is increasingly being recognized as a biophysical process involved in asymmetric cell division, for example the segregation of P granules in the early division of *C. elegans* embryos. The implication of LLPS in the asymmetric segregation of Numb is thus a timely topic and of general interest to people working in the field of asymmetric cell division. Overall the studies are carefully performed and the results are of high quality. There are a few comments the authors may want to consider to make this a stronger manuscript suitable for *Nat Commun*.

1) Overall, the *in vivo* evidence of LLPS involvement in Numb asymmetric localization of weak. Cultured HeLa cells and colocalization of Numb and Pon in the nucleus are used as the readout. How about the Numb-Pon interaction at the cell cortex of dividing NBs, which is the more relevant physiological setting?

*As the referee suggested, monitoring the Numb-Pon interaction-mediated phase transition at the basal cortex of dividing NBs, and testing whether disruption of this phase transition leads to defect of Numb localization would be the ideal in vivo assay to investigate the role of LLPS in mediating polarized protein condensation. The problem is that, unlike the LLPS-mediated assemblies of various cellular bodies or RNA-enriched granules, which are often round liquid droplet-like apartments stably existing within the cell that could be easily differentiated, the basal Numb-Pon assemblies are transiently formed, attached to the cortex and “pulled” into a crescent shape at one pole of the dividing cell. Formation of the apparently crescent shape may contain two processes: Numb-Pon interaction-induced phase separation and local complex condensation, and basal cortex anchoring, and the latter is most likely accompanied with mechanical pressure-induced shape change from the ball shape to the cap shape (crescent from the side view). These two processes could be sequential, or mixed with each other. Due to this complicated situation, it would be extremely difficult to monitor LLPS-induced Numb basal localization in dividing NBs. Similarly, it is very difficult to prove the LLPS-mediated postsynaptic densities formation (Zeng, *Cell*, 2016) and multivalent signaling pathways beneath the membrane (Li, *Nature*, 2012; Su, *Science*, 2016) in vivo.*

*At current stage, the well-recognized phenomenon to reflect in vivo LLPS processes is the fast, dynamic exchange of the components between condensed lipid phase and surroundings, as revealed by the FRAP analysis. Our LLPS-mediated Numb/Pon basal condensation model provides an ideal mechanistic explanation for the observation that Numb and Pon are in fast equilibrium between cortex crescent and cytoplasm in asymmetrically dividing *Drosophila* NBs and SOP cells (Lu, *Mol Cell*, 1999; Mayer, *Curr Biol*, 2005), as well as the stable existence of large concentration gradients of the proteins within the crescent and those in the cytoplasm. Combining our biochemical, cellular and in vivo fly assays, it is probably safe for us to propose that Numb-Pon interaction-induced LLPS is the driving force for their basal*

condensation.

2) Aliphatic alcohols, for example 1, 6 hexanediol has been used to disturb labial, phase-separating assemblies. Can the PON-Numb assemblies in vitro and in vivo be melted by Aliphatic alcohols?

Following the reviewer's suggestion, we analyzed the effect of 1,6-Hexanediol on the condensed Pon-Numb assemblies both in vitro and in vivo. As expected, preformed condensed phase droplets can be reversed by addition of 1,6-Hexanediol in a dose dependent manner (the updated Supplementary Fig 6, attached bellow for your reference). Moreover, in dividing NBs, treatment of 1,6-Hexanediol also disturbed the basal localization of both Numb and Pon in a dose dependent manner (the updated Supplementary Fig. 10a,c, attached bellow for your reference). Importantly, in the following recovery experiment using brain explant, removal of 1,6-Hexanediol restored the Numb/Pon basal crescent (Supplementary Fig. 10b,c), further demonstrating that the basal condensation of Numb/Pon is mediated by LLPS and highly dynamic.

Supplementary Figure 6 The Numb/Pon phase separation assemblies can be disturbed by 1,6-Hexanediol in vitro. (a) Pre-formed Numb PTB/Pon A1B3 (60 μ M) droplets are rapidly dispersed after adding 1,6-Hexanediol. The arrow refers to the time point of adding 1,6-Hexanediol to the mixture. (b) Pre-formed Numb PTB/Pon A1B3 (60 μ M) droplets can be reversed to aqueous phase by 1,6-Hexanediol in a dose dependent manner. The final concentration of 1,6-Hexanediol is indicated. All statistic data in this figure represent the results from three independent batches of experiments and are expressed as mean \pm SD.

Supplementary Figure 10 The basal localization of Numb/Pon can be disturbed by 1,6-Hexanediol in dividing NBs of larval brains. (a) In wild type NBs treated with 5% 1,6-Hexanediol, some (37.9%) NBs exhibit cytoplasmic localization of Pon and cortical localization Numb. When treated with 10% 1,6-Hexanediol, virtually all NBs show cytoplasmic Pon and cortical Numb. (b) After removal of 1,6-Hexanediol, the basal distribution of Numb/Pon is restored in NBs pre-treated with Hexanediol. ToPro-3 in white. White arrowheads point to basal cortex. Scale bars, 25 μ m for the whole brain and 5 μ m for NB images. (c) Statistical data for a and b.

3) Is it possible to use exogenous LLPS forming motifs from other proteins to replace the PON-Numb interacting motifs and rescue the Numb localization defect in Pon mutant. This would be the ultimate test of the role of LLPS in Numb localization.

The referee suggests a very good point. To do this experiment, we need to replace the AIB3 motif in Pon and the PTB domain in Numb with other LLPS forming motifs. And then, in type I pon mutant NBs, see whether these ectopically expressed chimeric Pon and Numb variants could retain their basal localization. A technical issue is that, our previous study suggested that the AI motif of Pon was also required for its basal localization. Cdk1 could phosphorylate T63 in the AI motif, which provides a docking site for Polo PBD domain, leads to PBD dimerization, finally results in Polo activation and subsequent Polo-mediated phosphorylation of Pon on S611 (Zhu, Structure, 2016). It has been demonstrated that S611 phosphorylation is essential for the basal localization of Pon (Wang, Nature, 2007). Thus, Pon AI motif may play regulated dual roles in mediating Numb basal localization. Moreover, the PTB domain is an important functional domain of Numb (e.g., recruiting Notch), which is also the only identified domain in Numb. Taking into consideration of the cell cycle dependent dynamic distribution of Numb, which is not always colocalized with Pon, removal of PTB may cause undesired effects (i.e. ectopic Notch signaling activation) which renders the experimental data difficult to be interpreted. Thus, this assay does not seem to be feasible, and we seek for this reviewer's understanding.

4) It is surprising that mouse Numb PTB domain does not interact with the PTB-binding motifs in Pon, although mammalian Numb has been shown to be asymmetrically localized in fly NSCs and rescue Numb mutant phenotypes. The authors should at least discuss this point.

We were also surprised to find that Pon A1B1, B1A2, A2B2, and A3B3 did not interact with mouse Numb PTB (Fig. 3d and Fig. IIa), and the full-length mouse Numb did not interact with Pon either (Fig. IIb). The localization of mouse Numb in Drosophila were investigated in the context of SOP during embryonic development (Zhong et al., 1996). Unlike its strict dependence on Pon for its localization in larval NB, the localization of Numb in embryonic SOP (and CNS) does not strictly depend on Pon (Lu et al., 1998). The asymmetric localization of Numb is only delayed in dividing SOP and still asymmetrically segregated into p11B as its wt counterpart, suggesting additional mechanism promotes asymmetric Numb localization in the absence of Pon. It is plausible that this Pon-independent “additional mechanism” is responsible for mouse Numb localization in embryonic SOP and rescue numb mutant SOP phenotypes.

Fig. II. Pon does not interact with mouse Numb. (a) ITC-based validation of the interaction between mouse Numb PTB and Pon fragments. (b) In HEK293T cells, Drosophila but not mouse Numb could be Co-IPed by Pon.

Reviewer #3:

This manuscript comprehensively investigated the interaction between the PTB domain of *Drosophila* Numb and one of its binding partner, Pon, through crystallographic, other biophysical and biochemical studies. They fully characterize the two binding surfaces of PTB: the canonical surface associating with one of three type A motifs and the other, a non-canonical surface, binding one of three type B motifs of Pon' s N-terminus. Interestingly such a bipartite binding mode on PTB not only ensures high specificity/affinity interaction between Numb and Pon but also, together with repeating linear motifs of Pon, enables multivalency-driven liquid-liquid phase transition (LLPT) by the two in vitro and in Hela cell. Furthermore, the authors show that perturbations (i.e. mutations of Numb or Pon) to LLPT also interfere with some essential functions of Numb and/or Pon, such as the control of Numb polarized location by Pon and the function of these two towards inhibiting Notch activities in fly.

If the following concerns are adequately addressed, I find this manuscript of the significance and general interest to be published in Nature Communication.

A major point:

Liquid-liquid phase transition of biomacromolecules is quickly emerging as a field of research in biology. This field was largely initiated by the combination of two landmark publications: one in Science in 2009 and the other in Nature in 2012. The latter paper is especially relevant here because it has explicitly established the multivalency-driven LLPT concept in biology. The manuscript cites a few recent publications while missing the two landmark papers, which are pretty recent as well.

We thank the reviewer for pointing out these missing references, and have cited them in the revised manuscript.

Minor points:

1) Figure 1b shows that the binding affinities between Numb PTB and Pon's di-motif constructs are consistently higher than those between PTB and Pon's tri-motif constructs. It hardly makes thermodynamic sense. It is likely due to a wrong model used for ITC data fitting. Therefore, a discussion of this apparent inconsistency is needed to avoid confusion. Alternatively, the latter set of ITC data shall be left out, which doesn't affect any of the major conclusions of the paper.

Following this reviewer's suggestion, we have removed the ITC data of Pon's tri-motif constructs.

2) The observed phase transition is in three-dimension in vitro and in Hela cell while the crescent shaped Numb/Pon complex is more or less in two-dimension in fly. A possible mechanism leading to dimension reduction has to be discussed. For example, a membrane associating domain within or a lipid modification on Numb or Pon or a third partner of the Numb/Pon complex is responsible for the apparent membrane attachment/localization.

We thank the reviewer for this useful suggestion, and have modified the manuscript.

REVIEWERS' COMMENTS:

Reviewer #2 (Remarks to the Author):

The authors have adequately addressed my concerns and as a result the manuscript has been significantly improved. It is now recommended for publication in Nature Communications.

(Our responses to the reviewers' comments are shown in italics and highlighted in blue):

Reviewer #2 (Remarks to the Author):

The authors have adequately addressed my concerns and as a result the manuscript has been significantly improved. It is now recommended for publication in Nature Communications.

We thank Reviewer #2 for his/her final approval.